# Organic hydrogen peroxide-driven low charge potentials for high-performance lithium-oxygen batteries with carbon cathodes

Shichao Wu[1,2], Yu Qiao[1,2], Sixie Yang[3], Masayoshi Ishida[2], Ping He[3] & Haoshen Zhou[1,2,3]

Reducing the high charge potential is a crucial concern in advancing the performance of lithium-oxygen batteries. Here, for water-containing lithium-oxygen batteries with lithium hydroxide products, we find that a hydrogen peroxide aqueous solution added in the electrolyte can effectively promote the decomposition of lithium hydroxide compounds at the ultralow charge potential on a catalyst-free Ketjen Black-based cathode. Furthermore, for non-aqueous lithium-oxygen batteries with lithium peroxide products, we introduce a urea hydrogen peroxide, chelating hydrogen peroxide without any water in the organic, as an electrolyte additive in lithium-oxygen batteries with a lithium metal anode and succeed in the realization of the low charge potential of $\sim 3.26$ V, which is among the best levels reported. In addition, the undesired water generally accompanying hydrogen peroxide solutions is circumvented to protect the lithium metal anode and ensure good battery cycling stability. Our results should provide illuminating insights into approaches to enhancing lithium-oxygen batteries.

[1] Energy Technology Research Institute, National Institute of Advanced Industrial Science and Technology (AIST), 1-1-1, Umezono, Tsukuba 305-8568, Japan. [2] Graduate School of System and Information Engineering, University of Tsukuba, 1-1-1, Tennoudai, Tsukuba 305-8573, Japan. [3] Center of Energy Storage Materials and Technology, College of Engineering and Applied Sciences, National Laboratory of Solid State Microstructures, Collaborative Innovation Center of Advanced Microstructures, Nanjing University, Nanjing 210093, China. Correspondence and requests for materials should be addressed to H.Z. (email: hs.zhou@aist.go.jp).

Rechargeable Li-air batteries with the high energy density of $\sim 3,500$ Wh kg$^{-1}$ demonstrate great promise for building capacious energy storage devices and developing electric vehicles with long driving range[1]. A typical Li-air battery is composed of a Li metal anode, a separator, an electrolyte and an air cathode[2]. The general net electrochemical reactions of non-aqueous Li-O$_2$ batteries during discharge and charge are based on the formation and oxidative decomposition of lithium peroxide ($2\,Li + O_2 \rightleftarrows Li_2O_2$, $E^0 = 2.96$ V versus Li/Li$^+$)[3–5]. The insulating nature of Li$_2$O$_2$ depositing on the surface of the air cathode may result in sudden battery death or limited discharge capacity during discharge[6–9]. Upon charge, the potential will be too high to decompose Li$_2$O$_2$ ($>4.2$ V)[10]. This may trigger the severe oxidative deterioration of the electrolyte and shorten the cycle life of the battery[11,12]. Such problems have blocked the realization of practical Li-air batteries. Carbon-based materials have been considered to be optimal selections for air cathodes, because their superior conductivity can buffer the poor electron conductivity of Li$_2$O$_2$ and thereby improve the discharge–charge performance[13,14]. They also show other advantages such as low cost and high specific area and light weight, which ensure the high specific capacity and energy density of Li-O$_2$ batteries[15]. However, their poor catalysis ability towards the oxygen evolution reaction restrains the charge potential from realizing low values[14]. Substantial efforts have been devoted to lowering the charge potential[16–21]. Noble metals and oxides (Pt, Au, Pd, Ru and RuO$_2$ and so on)[22–27] and transition metals and compounds (MnO$_2$, TiC, Ti$_4$O$_7$, Cu$_2$O, FeOOH, NiOOH, Ni$_2$CoO$_4$ and so on) have been tentatively introduced to construct carbon-based composite cathodes[28–30]. In this way, charge potentials of $\sim 3.5$ V can be anticipated, but it is difficult to break through this limit. Also of note, the employment of inappropriate catalysts that strongly bind O$_2$ or the discharge intermediate LiO$_2$ may result in the undesired shift from Li$_2$O$_2$ to Li$_2$O as discharge products and further lead to the poor reversibility of the Li-O$_2$ battery[31,32].

Recently, Li et al.[33] in our group demonstrated a novel route to achieve the ultralow charge potential of $\sim 3.2$ V in a dimethyl sulfoxide-based electrolyte containing 100 p.p.m. of H$_2$O by constructing a composite cathode (MnO$_2$ and Ru particles supported on Super P carbon). The primary electrochemical reactions (the formation of Li$_2$O$_2$) during discharge were converted to the formation of lithium hydroxide (LiOH) via the catalysis effect of MnO$_2$ in the cathode towards the reactions between Li$_2$O$_2$ and H$_2$O. The discharge process was proposed to involve a 2e$^-$ electrochemical reaction and a following chemical reaction. During charge, Ru in the cathode decomposes the LiOH at the low charge potential. Following this idea, in a tetraglyme (G4)-based electrolyte, 4,600 p.p.m. of H$_2$O was introduced to reduce the charge potential to $\sim 3.3$ V[34] and, by integrating a hydrophobic ionic liquid-based electrolytes, Wu et al.[35] realized a synergistic system for Li-O$_2$ batteries in a humid atmosphere (relative humidity of 51%) and a charge potential of $\sim 3.34$ V was attained. Although the transformed net electrochemical reactions based on the reversible formation and decomposition of LiOH compounds efficiently improve the charge ability and cycling performance, the introduction of MnO$_2$ and Ru or RuO$_2$, with their heavy molecular weight and high cost, such as the abovementioned carbon-based composite cathodes, is inevitably subject to significantly decreased energy density and increased cost. Moreover, the overly strong catalytic activity of these additional catalysts may lead to parasitic reactions of the electrolyte. In the cases with carbon cathodes containing no catalysts, the charge potentials have been very high, generally above 4.2 V, although H$_2$O was reported to largely enhance the discharge capacity[7,36,37]. Another work by Liu et al.[38], also based on the assumed LiOH-related discharge–charge mechanism,

reported a charge potential of $\sim 3.1$ V on an reduced graphene oxide (rGO)-based cathode in a dimethyl ether-based electrolyte in the presence of LiI and H$_2$O. However, these features were not observed for Li-O$_2$ batteries with common carbon material-based cathodes such as Super P carbon. Furthermore, some discussion challenged the detailed function of LiI and the reversibility of the Li-O$_2$ battery with LiOH as the discharge product and suggested that the uncertain mechanism should be further explored[39–42].

In this work, for an H$_2$O-containing Li-O$_2$ battery with LiOH products, we reveal the critical role of an H$_2$O$_2$ aqueous solution added into the electrolyte in assisting the decomposition of LiOH compounds at the ultralow charge potential when using a common Ketjen Black carbon (KB)-based cathode. Most importantly, for non-aqueous Li-O$_2$ batteries with Li$_2$O$_2$ products, a novel electrolyte additive urea hydrogen peroxide that traps H$_2$O$_2$ in the H$_2$O-free organic is, to the best of our knowledge, first introduced to reduce the charge potential to as low as $\sim 3.26$ V and simultaneously avoid Li metal anode corrosion by H$_2$O.

## Results

**Low charge potential for decomposing LiOH compounds.** As H$_2$O$_2$ generally exists in aqueous solution (30 wt%), to study the charge behaviour for decomposing LiOH compounds with or without the presence of H$_2$O$_2$ aqueous solution in the electrolyte, a Li metal anode-protected pouch cell is fabricated to prevent side reactions between the Li metal anode and H$_2$O in the electrolyte[43]. On the anode side, the Li metal is protected by utilizing a Li ion-conducting glass-ceramic film (LiSICON, Li$_2$O-Al$_2$O$_3$-SiO$_2$-P$_2$O$_5$-TiO$_2$-GeO$_2$, Ohara Corporation, Japan), only allowing the transport of Li$^+$ and preventing the penetration of other ions. On the cathode side, a KB-based cathode and a glass fibre infiltrating electrolyte are constructed. Figure 1a presents the charge profiles for decomposing solid LiOH compounds on the KB-based cathode in the electrolytes with and without H$_2$O$_2$ solution. The KB-based cathodes in situ loaded with solid LiOH compounds were obtained by first discharging Li-O$_2$ pouch cells in the dry electrolyte to 1.50 mAh (corresponding to $\sim 4,000$ mAh g$^{-1}$$_{KB}$ and 3.5 mAh cm$^{-2}$, Fig. 1a,i), with Li$_2$O$_2$ as the product being evidenced by the X-ray diffraction (XRD) pattern in Fig. 1c. Then, the discharged cathodes were extracted in an Ar glove box and left in an Ar atmosphere with a relative humidity of 75% for 7 days. The XRD pattern in Fig. 1d confirms that all the Li$_2$O$_2$ on the discharged cathode was converted to a mixture of LiOH and LiOH·H$_2$O. New Li-O$_2$ pouch cells (Fig. 1a,ii) were assembled with these cathodes and H$_2$O$_2$-containing/free electrolytes to investigate the effect of H$_2$O$_2$ on the charge performance. In the G4-H$_2$O electrolyte, the charge potential increased rapidly to a plateau at $\sim 4.16$ V, corresponding to the overpotential of 1.20 V. This high value indicates the difficulty of decomposing LiOH compounds at the KB-based cathode without an efficient catalyst. At the end of the charge process (Fig. 1a,iii), the charge capacity is only $\sim 0.6$ mAh, much lower than the discharged capacity ($\sim 1.50$ mAh). The XRD pattern of the recharged cathode in Fig. 1e,iii illustrates that there are undecomposed LiOH compounds. We rationalize these observations by the poor catalytic activity of KB and the weakening contacts between KB particles and LiOH compound particles during the charging process[33]. In contrast, when H$_2$O$_2$ aqueous solution is introduced into the electrolyte, the charge plateau is greatly decreased to $\sim 3.50$ V and the charge capacity is increased to above 1.40 mAh. It should be noted that 3.50 V is quite low, considering the large electrochemical impedance of the employed LiSICON film. The low charge overpotential of $\sim 0.54$ V and the high recharge

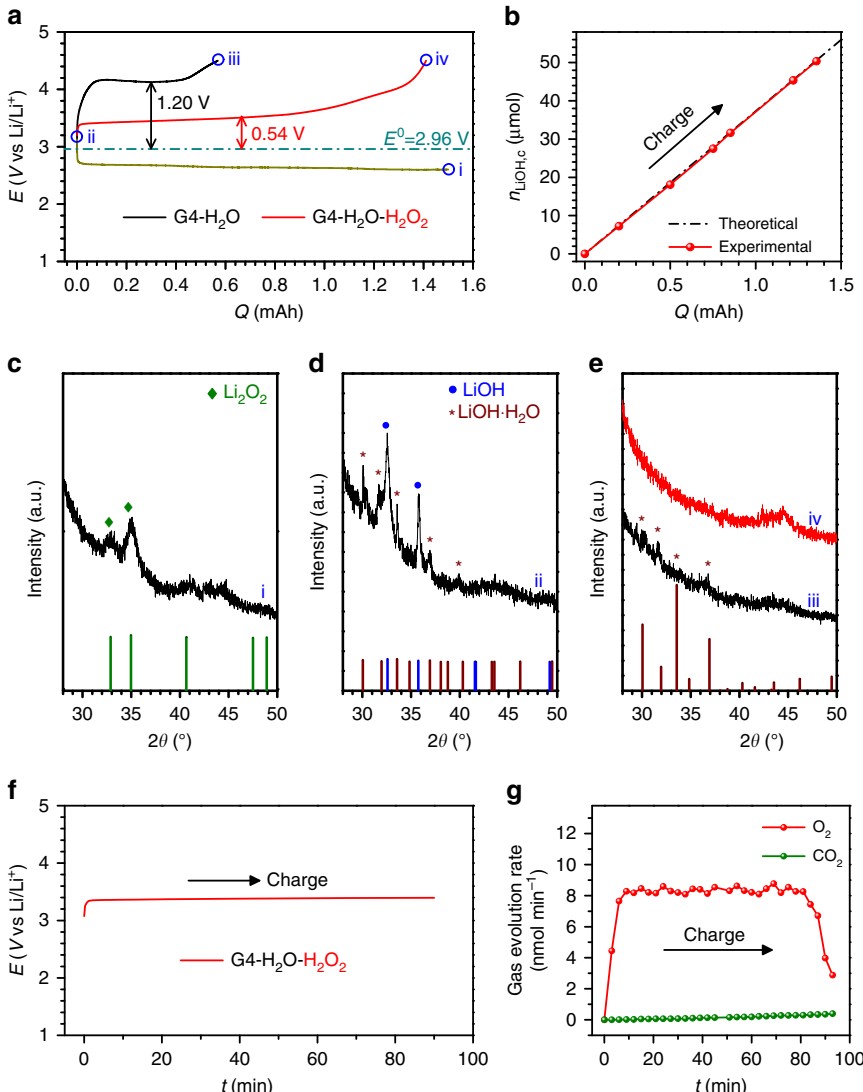

**Figure 1 | The reduced charge potentials for decomposing LiOH compounds formed *in situ*.** (**a**) The discharge and charge profiles at 100 mA g$^{-1}_{KB}$. During charging, solid LiOH compounds can be oxidatively decomposed at a low potential of 3.50 V (corresponding to the overpotential of 0.54 V) in the presence of $H_2O_2$ solution in the electrolyte. This is much lower than the value (1.20 V) without $H_2O_2$. (**b**) The quantity of LiOH compounds consumed, $n_{LiOH,c}$, during charge determined by titration. (**c**–**e**) XRD patterns of cathodes corresponding to the different states (i, ii, iii and iv) in **a**. The Li-O$_2$ pouch cell is first discharged to 1.5 mAh (corresponding to $\sim$4,000 mAh g$^{-1}_{KB}$ and 3.5 mAh cm$^{-2}$) in the dry electrolyte to produce $Li_2O_2$ at the cathode (**c**). The $Li_2O_2$ is converted to a mixture of LiOH and LiOH·$H_2O$ (**d**) by keeping the cathode (i) in an Ar atmosphere with a relative humidity of 75% for 7 days. After charging, the LiOH and LiOH·$H_2O$ are reversibly oxidized at low charge potential in the presence of $H_2O_2$ in the electrolyte, evidenced by the disappearance of their diffraction peaks in **e**,iv, whereas LiOH·$H_2O$ remains undecomposed in the absence of $H_2O_2$ in the electrolyte (**e**,iii). (**f**) Charge profiles of Li-O$_2$ pouch cell with *in situ* formed solid LiOH/KB cathode and G4-$H_2O$-$H_2O_2$ electrolyte during the in situ DEMS measurement. (**g**) Corresponding O$_2$ and CO$_2$ evolution rates during charge. Current density: 100 mA g$^{-1}_{KB}$. A LiSICON film preventing $H_2O$ penetration was employed to fabricate the Li-O$_2$ pouch cells.

Coulombic efficiency of above 93% indicate the exceptional benefit of $H_2O_2$ in the electrolyte for decomposing LiOH compounds. The disappearance of LiOH compound peaks after recharge (Fig. 1e,iv) confirms their full decomposition and good charge reversibility. After charge, the electrolyte exhibits no obvious decomposition and no formation of soluble species, as evidenced by Fourier transform infrared (FTIR) spectra (Supplementary Fig. 1).

To quantitatively examine the charge process corresponding to the LiOH oxidation in the $H_2O_2$-containing electrolyte, we conducted a titration experiment to determine the consumed amount of LiOH at various charge states. The amount of these LiOH compounds ($n_{LiOHii}$) was quantified by titration following

the method of McCloskey *et al.*[44] When the newly assembled cells were charged to certain capacities (0.2, 0.5, 0.75, 0.85, 1.22 or 1.356 mAh), the residual amount of LiOH compounds was titrated and the consumed amount of LiOH compounds could be obtained by subtracting the LiOH compounds at each point from $n_{LiOHii}$. As shown in Fig. 1b, at each point during the charge process, the calculated amount of LiOH compounds consumed was almost equal to the theoretical value and the relationship of the consumed amount and the charge capacity is linearly dependent. *In situ* differential quantitative mass spectrometry (DEMS) was performed to examine whether O$_2$ could be generated from LiOH decomposition at the low charge potential in the presence of $H_2O_2$. As shown in Fig. 1f,g, clear

$O_2$ evolution can be detected at $\sim 3.50$ V during the charge process and there is no evolution of $CO_2$. Accordingly, these quantitative results indicate that the oxidation of LiOH compounds dominates in the charge process.

We also examined the charge performance of commercial LiOH-preloaded KB-based cathodes in the $H_2O_2$-containing/free electrolytes. The results are shown in Fig. 2a and Supplementary Fig. 2. With the aid of $H_2O_2$ in the electrolyte, the preloaded LiOH in the KB-based cathode is much easier to fully decompose at a low charge potential ($\sim 3.61$ V, Fig. 2a) than in the absence of $H_2O_2$ ($> 4.40$ V). In Fig. 2b, we present the charge profiles of Li-$O_2$ pouch cells with dissolved LiOH aqueous solution and dissolved LiOH-$H_2O_2$ aqueous solution in the electrolytes. The presence of $H_2O_2$ allows a large reduction of charge overpotentials to 0.55 V from 1.50 V in the absence of $H_2O_2$. After charge at high potential in G4-$H_2O$-LiOH$_{(l)}$ electrolyte, the LiOH peak at $\sim 3,680$ cm$^{-1}$ in FTIR spectra (Supplementary Fig. 3,iii) remains, indicating the uncomplete decomposition of liquid LiOH in the electrolyte. In contrast, the LiOH can be fully decomposed after charge at low potential in the $H_2O_2$-containing electrolyte, as evidenced by the peak having disappeared (Supplementary Fig. 3,iv). This phenomenon may provide some insights into the improvement of the aqueous Li-$O_2$ battery by adding $H_2O_2$ into the aqueous electrolyte; further work will continue to explore this phenomenon. All of these results emphasize that the introduction of $H_2O_2$ in the electrolyte can greatly enhance the decomposition ability of either solid LiOH compounds at common carbon cathodes or liquid LiOH in the electrolyte at low potentials during the charge process.

In view of the superiority of $H_2O_2$ in improving the charge performance in Li-$O_2$ batteries, the cycling stability is expected to be promoted. Supplementary Fig. 4 shows the discharge–charge profiles of Li-$O_2$ pouch cells with $H_2O_2$ aqueous solution in the electrolyte. Pure KB is used as the active material for the cathode and LiSICON film is used to protect the Li metal anode from corrosion by $H_2O$. The pouch cells were discharged and charged with a limited specific capacity of 1,000 mAh g$^{-1}_{KB}$ at 100 mA g$^{-1}_{KB}$ and a voltage range of 2.00–4.50 V. In the ten cycles, the discharge plateaus are at $\sim 2.70$ V and remain nearly unchanged. For the charge profiles, the first plateau is at $\sim 3.45$ V and the charge-specific capacity can reach 1,000 mAh g$^{-1}_{KB}$ with a terminal charge potential of $< 4.00$ V at the end of charge process, indicating the strong charge ability and reversibility. In the initial five cycles, the low charge plateaus remain and the terminal charge potentials show no notable increase. Until the sixth cycle, the charge plateau increases to $\sim 3.65$ V and the terminal potential reaches the limitation of 4.50 V in addition to the charge-specific capacity decreasing to 950 mAh g$^{-1}_{KB}$. After ten cycles, the charge profile shows a large increment (to $\sim 4.00$ V) of the plateau and poor reversibility (charge capacity of only 800 mAh g$^{-1}_{KB}$). Electrochemical impedance analysis was performed before and after the cycles to determine the cause of the fading cycling stability (Supplementary Fig. 5). The first arcs in the high-frequency region corresponding to the interphase impedance are almost constant before and after ten cycles. However, the second arc in the middle-frequency region corresponding to the charge transfer notably changes[45,46]. The large increase from 2.5 k$\Omega$ in the first cycle to 3.6 k$\Omega$ after the ten cycles is responsible for the difficulty of the Li$^+$ transfer through the LiSICON film and the resulting poor cycling performance. These results imply that adopting the LiSICON film to overcome the $H_2O$ addition accompanying the $H_2O_2$ aqueous solution in the Li-$O_2$ pouch cell is inadvisable in terms of realizing long cycle life and alternative strategies should be sought to exploit the merits of $H_2O_2$ for enhancing the performance of Li-$O_2$ batteries.

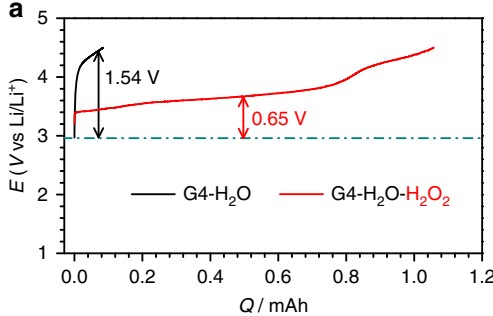

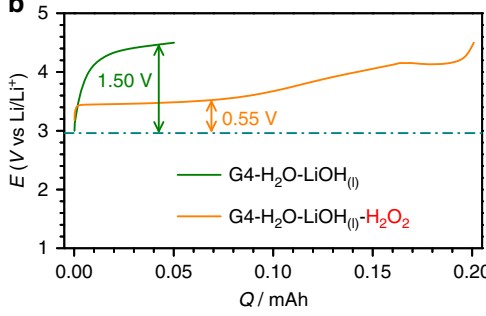

**Figure 2 | The reduced charge potentials for decomposing preloaded LiOH.** (**a**) Preloaded commercial LiOH on a KB-based cathode at 100 mA g$^{-1}_{KB}$ and (**b**) liquid LiOH in the G4-based electrolyte by the action of $H_2O_2$ (current density: 100 mA g$^{-1}_{KB}$). During charging, the preloaded LiOH can be decomposed at a low potential of $\sim 3.61$ V (corresponding to the overpotential of 0.65 V) in the presence of $H_2O_2$ solution in the electrolyte. This is much lower than the value (1.54 V) without $H_2O_2$. For the liquid LiOH, the charge overpotential is reduced from 1.50 V in the absence of $H_2O_2$ to 0.55 V in the presence of $H_2O_2$ in the electrolyte. The battery configuration is that of the Li-$O_2$ pouch cell.

**Organic $H_2O_2$ compound introduction.** In the general battery configuration with a KB-based cathode/glass fibre separator infiltrating electrolyte/Li metal anode structure, if the normal $H_2O_2$ aqueous solution is introduced in the electrolyte (Fig. 3a), the concomitant $H_2O$ will inevitably attack the Li metal anode, causing battery death and possible safety issues. As shown in Fig. 3d, even only 5,000 p.p.m. of $H_2O$ in the electrolyte can lead to serious corrosion of the Li metal anode after cycling. Large amounts of LiOH (determined by the XRD, Fig. 3c) cover the surface of the Li metal anode and cut off the Li$^+$ generation and transfer. This would probably result in the decreased discharge-specific capacity and battery failure within only a few cycles (Fig. 3b).

To exploit the merits of $H_2O_2$ for reducing the charge overpotential and extending cycle life, introducing $H_2O_2$ in the electrolyte while eliminating $H_2O$ contamination should be a quick fix. Herein, we pilot an organic $H_2O_2$ compound (urea hydrogen peroxide, UH$_2O_2$) as the electrolyte additive to lower the charge potential and more importantly, to prohibit $H_2O$-related side reactions (Fig. 4a). Urea has been reported to form deep eutectic electrolytes with LiTFSI, have pronounced effects on the transport and structural properties of LiTFSI and induce faster Li$^+$ diffusion and improved ion transport[47]. The ion conductivity of the G4-UH$_2O_2$ electrolyte is determined to $1.8 \times 10^{-3}$ S cm$^{-1}$ through the Nyquist plot in Supplementary Fig. 6. This value is comparable to the levels of commercial electrolytes for Li ion batteries and indicates the high Li$^+$ conductivity. In the electrolyte containing UH$_2O_2$, the rate

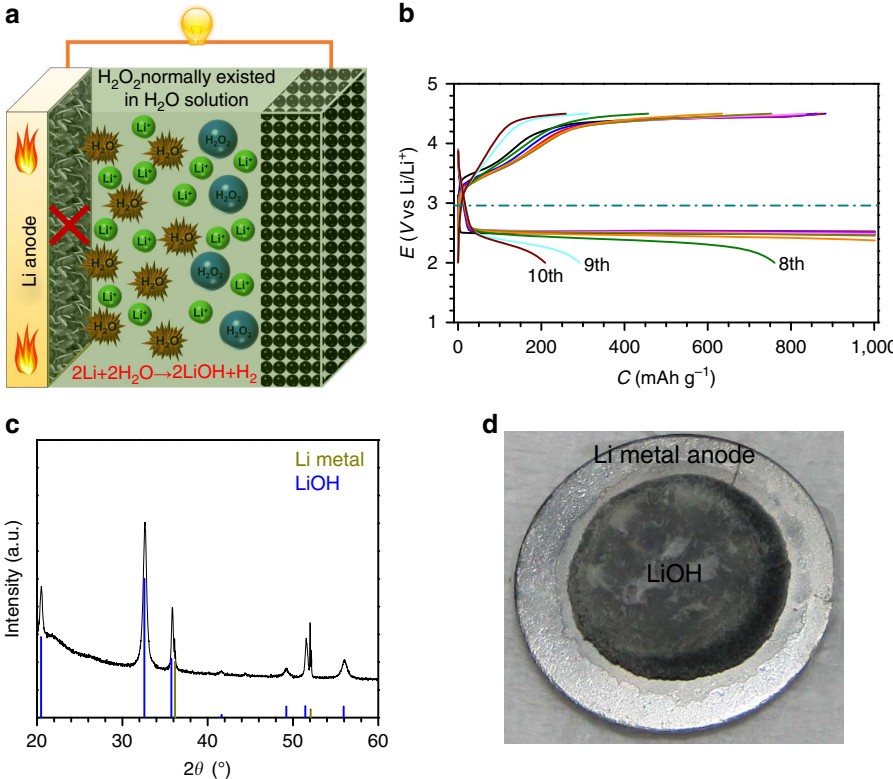

**Figure 3 | $H_2O$-induced poor battery performance and corrosion of the Li metal anode.** (**a**) Schematic illustrations of Li-$O_2$ cells with a general configuration (KB-based cathode/glass fibre separator infiltrating electrolyte/Li metal anode) in the G4-$H_2O$-$H_2O_2$ electrolyte. (**b**) Discharge–charge profiles of Li-$O_2$ cells with 5,000 p.p.m. of $H_2O$ in the electrolyte (current density: 500 mA g$^{-1}$$_{KB}$). (**c**) The XRD pattern of the Li metal anode after cycling in the electrolyte containing 5,000 p.p.m. of $H_2O$. (**d**) A photo of a Li metal anode after cycling in the electrolyte containing 5,000 p.p.m. of $H_2O$. In the presence of $H_2O_2$ aqueous solution, the Li metal anode was significantly corroded, resulting in LiOH on the surface, and the battery performance faded rapidly.

performances of Li-$O_2$ cells were evaluated at current densities of 100, 250, 500 and 1,000 mA g$^{-1}$$_{KB}$; the results are given in Fig. 4b. At 100 mA g$^{-1}$$_{KB}$, the charge plateau is at $\sim$3.26 V, corresponding to the much lower overpotential of 0.30 V, which is among the lowest values reported. Even at the high current density of 1,000 mA g$^{-1}$$_{KB}$, the charge plateau is only $\sim$3.44 V, suggesting the excellent rate capability. With regard to the Li metal anode, no LiOH was detected on the surface of the Li metal anode after cycling, as expected (Fig. 4c). The silvery grey colour in the photo in Fig. 4d confirms the Li metal anode to be intact, in sharp contrast with the almost damaged Li metal anode (photo in Fig. 3d). In addition, from the storage experiments (Supplementary Fig. 7), the steady open circuit voltage trends after a discharge process or one discharge–charge cycle indicate the solid stability of the Li-$O_2$ cells with the G4-U$H_2O_2$ electrolyte. The results verify that employing the organic U$H_2O_2$ as an electrolyte additive is an accessible way to ensure the attractive charge performance of the Li-$O_2$ cell and achieve high stability of the Li metal anode, along with maintaining reliable safety.

The discharge products on the cathode of the Li-$O_2$ cell in the G4-U$H_2O_2$ electrolyte were analysed by XRD and scanning electron microscopy (SEM). Figure 5a summarizes the XRD patterns of pristine, discharged and recharged cathodes. The discharge products are determined to be Li$_2O_2$. After recharge, the diffraction peaks of Li$_2O_2$ disappear, implying its full decomposition. The typical toroid morphology of Li$_2O_2$ after discharge can be directly observed from Fig. 5c. After the cell is recharged, the particles are completely decomposed (Fig. 5d). This is in good agreement with the XRD results.

As the reactions during discharge and charge in the Li-$O_2$ cell are complex multiphase processes, it is necessary to confirm whether the low charge potential in the Li-$O_2$ cell with the G4-U$H_2O_2$ electrolyte resulted from the improved charge ability or the unexpected side reactions. Therefore, we conducted in situ DEMS measurement to monitor the $O_2$ evolution during charge and titration experiments, to quantitatively analyse the Li$_2O_2$ formation and consumption during discharge and charge in the Li-$O_2$ cell with the G4-U$H_2O_2$ electrolyte. The corresponding data are provided in Fig. 6. After discharging the cell to 0.1, 0.2, 0.3, 0.4 or 0.5 mAh at 0.05 mA, the Li$_2O_2$ formation was quantified by a titration method (Fig. 6c). The yields of the formed Li$_2O_2$ (the amount of Li$_2O_2$ titrated divided by the amount of Li$_2O_2$ expected given the coulometry) held at $\sim$91%. This value is similar to the yield reported by McCloskey et al.[44] and indicates some slight side reactions, possibly corresponding to inevitable electrolyte decomposition. The value of e$^-$/Li$_2O_2$ can be estimated to be 2.15, close to the theoretical value of 2. The gas evolution rates of $O_2$ and $CO_2$ upon charge are presented in Fig. 6b. $O_2$ evolution was detected from the beginning and continued along the charge potential of $\sim$3.3 V during the whole charge process. No evolution of $CO_2$ was detected. The amounts of Li$_2O_2$ consumed during charge were analysed by titration measurement (Fig. 6d). Li$_2O_2$ oxidation follows an $\sim$2.14 e$^-$/Li$_2O_2$ process during the whole charge process. This value approaches the ideal value of 2 and clearly demonstrates that the significant reduction of charge overpotential due to the U$H_2O_2$ additive in the electrolyte can prohibit the high charge potential-induced side reaction and that the introduction of U$H_2O_2$ can improve the charge performance. The DEMS and

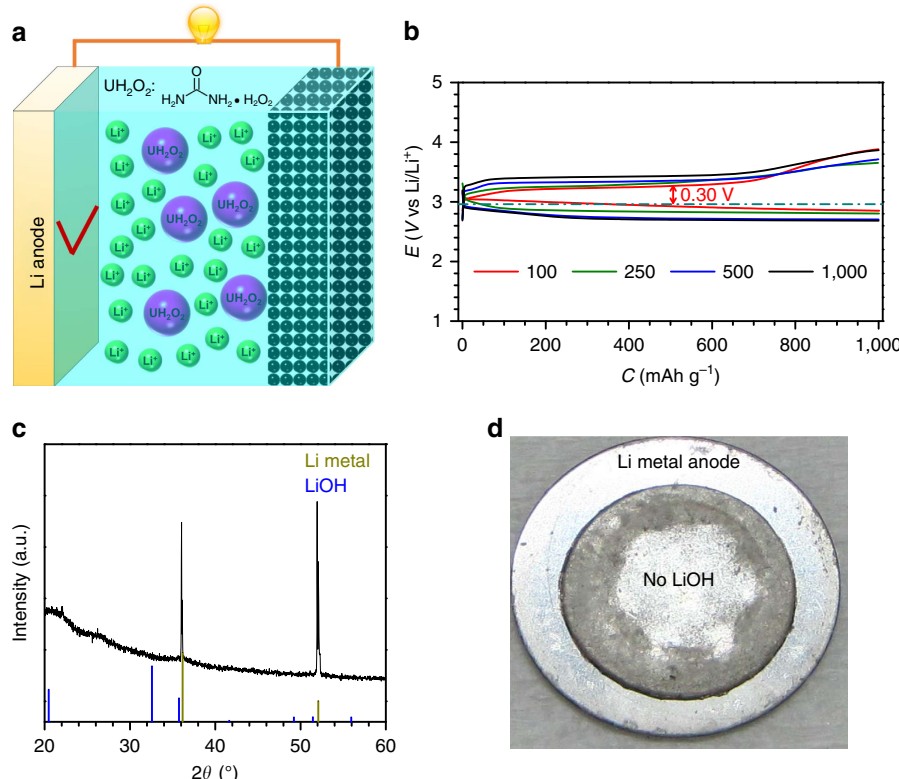

**Figure 4 | Blocking of $H_2O$-induced problems in the developed G4-$UH_2O_2$ electrolyte.** (**a**) Schematic illustrations of Li-$O_2$ cells with a general configuration (KB-based cathode/glass fibre separator infiltrating electrolyte/Li metal anode) in the developed G4-$UH_2O_2$ electrolyte. (**b**) Rate performance of a Li-$O_2$ coin cell in the G4-$UH_2O_2$ electrolyte. (**c**) The XRD pattern of the Li metal anode after cycling in the G4-$UH_2O_2$ electrolyte. (**d**) The photo of a Li metal anode after cycling in the G4-$UH_2O_2$ electrolyte. In the G4-$UH_2O_2$ electrolyte without $H_2O$, the Li metal anode remained undamaged. The Li-$O_2$ coin cell achieves the ultralow charge potential of 3.26 V (overpotential of 0.30 V) at 100 mA g$^{-1}$$_{KB}$ with an excellent rate capability.

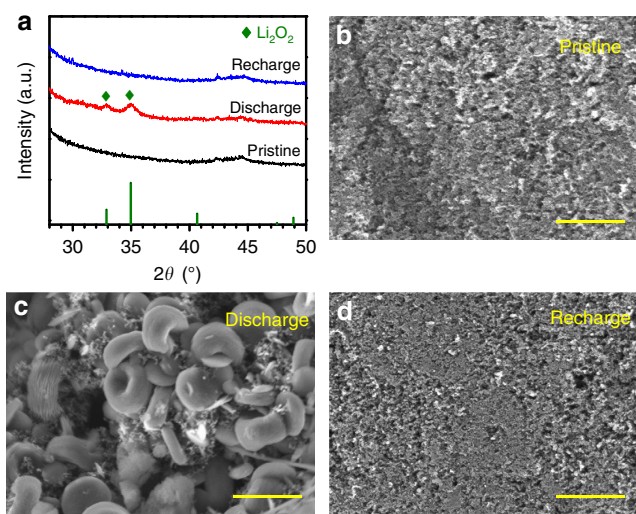

**Figure 5 | Discharge products on the cathode of a Li-$O_2$ cell in the G4-$UH_2O_2$ electrolyte.** (**a**) XRD patterns of the pristine, discharged and recharged cathodes. SEM images of (**b**) the pristine cathode, (**c**) the discharged cathode and (**d**) the recharged cathode. The discharge products are identified as $Li_2O_2$ with typical toroid morphology. Scale bar, 2 μm.

titration results suggest that the charge reactions are dominated by the evolution of $O_2$ and the consumption of $Li_2O_2$. Therefore, the high rechargeability of the Li-$O_2$ cell with the G4-$UH_2O_2$ electrolyte can be affirmed.

A comparison of the discharge–charge profiles of Li-$O_2$ cells in the electrolytes with and without $UH_2O_2$ in Fig. 7a,b, respectively,

confirms the critical role of $UH_2O_2$ in ameliorating the insufficient oxygen evolution reaction catalysis ability when adopting pure KB carbon as the air cathode. At 500 mA g$^{-1}$$_{KB}$, the Li-$O_2$ cell based on the KB cathode in the $UH_2O_2$-free electrolyte shows a charge potential as high as ∼4.30 V. This high value can result in the decomposition of electrolyte and the short cycle life of the Li-$O_2$ cell. After ∼20 cycles, the charge-specific capacity cannot recover to 1,000 mAh g$^{-1}$$_{KB}$ with a decreased Coulombic efficiency of ∼70%. With respect to the G4-$UH_2O_2$ electrolyte, the cycling performance of Li-$O_2$ cell is shown in Fig. 7a,c. The electrolyte stability was evaluated through FTIR analysis (Supplementary Fig. 8). Compared with the pristine G4-$UH_2O_2$ electrolyte (iii), there is nearly no change in the FTIR spectra of the G4-$UH_2O_2$ electrolytes after discharge (iv) and charge (v), indicating no obvious decomposition of the electrolyte. After 50 cycles, the discharge profiles with plateaus at ∼2.68 V show almost no change, indicating good stability. The charge profiles present low plateaus at ∼3.33 V in the initial cycles, with a slight increment after 50 cycles. All of the 50 charge-specific capacities can achieve 1,000 mAh g$^{-1}$$_{KB}$ with a Coulombic efficiency of 100%, implying excellent reversibility. The terminal charge potentials are restricted below ∼4.20 V. The improved cycling stability is attributed to this developed electrolyte additive $UH_2O_2$.

Considering the efficient assistance of $H_2O_2$ for decomposing the preloaded LiOH in the KB-based cathode (Fig. 2), $UH_2O_2$ is also expected to solve the high charge potential for decomposing the preloaded commercial $Li_2O_2$. It is an essential prerequisite for constructing Li metal-free Li ion-$O_2$ batteries[48]. Supplementary Fig. 9 presents the first charge profiles of Li-$O_2$ cells with commercial $Li_2O_2$-preloaded cathodes. In the electrolyte without

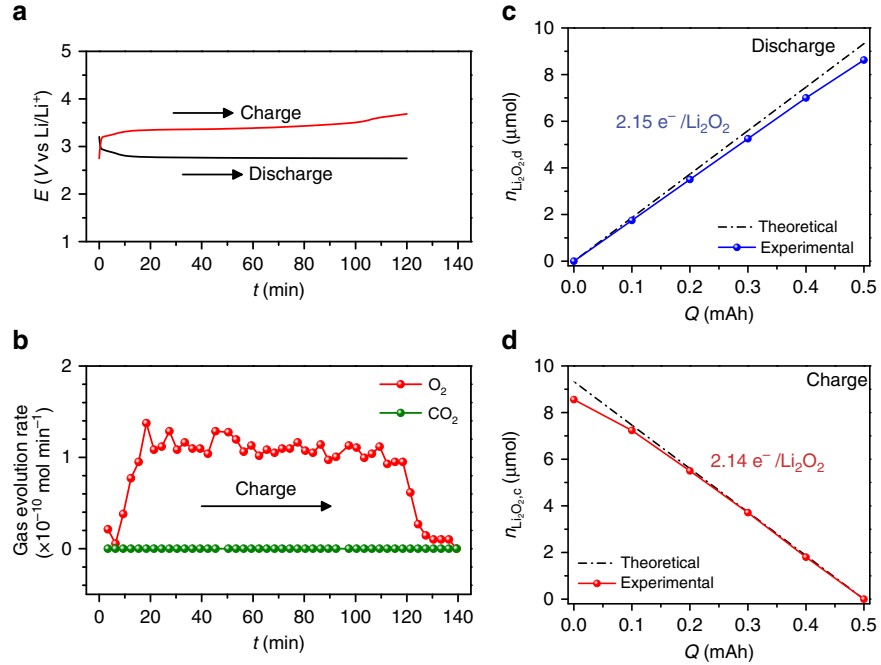

**Figure 6 | Reversibility during discharge and charge processes in the G4-UH₂O₂ electrolyte.** (**a**) Discharge and charge profiles of a Li-O₂ cell with G4-UH₂O₂ electrolyte during the in situ DEMS measurement. (**b**) Corresponding O₂ and CO₂ evolution rates during charge. (**c**) The quantified level of Li₂O₂ formation, $n_{Li_2O_2,d}$, during discharge determined by titration. (**d**) The quantified level of Li₂O₂ consumption, $n_{Li_2O_2,c}$, during charge determined by titration.

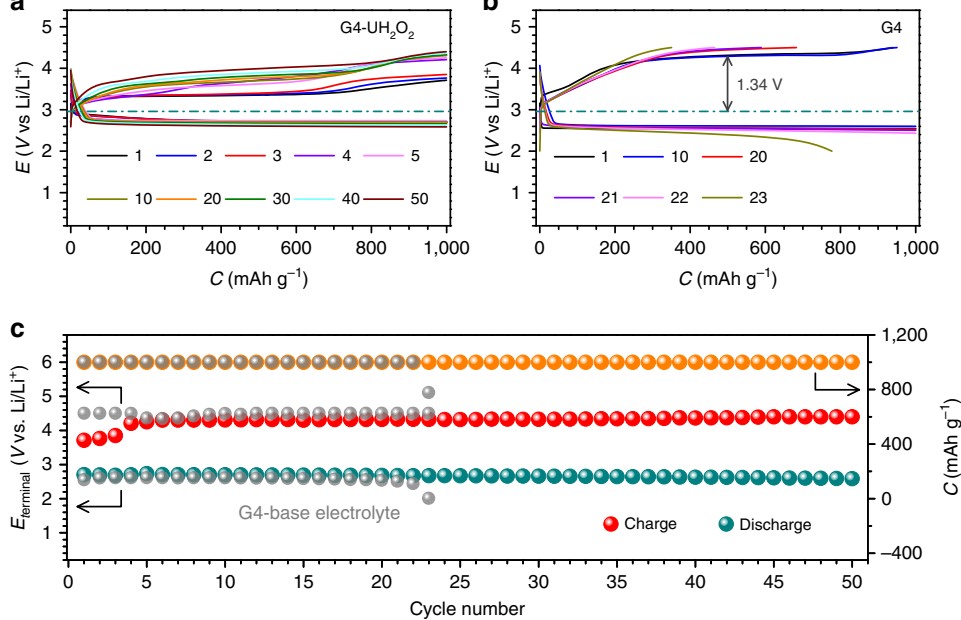

**Figure 7 | Cycling performance of the Li-O₂ cell.** Selected discharge and charge profiles in (**a**) the G4-UH₂O₂ and (**b**) G4-based electrolytes. (**c**) The corresponding discharge and charge terminal voltages along with capacities upon cycling. Current density: 500 mA g⁻¹_KB.

any additive, a high potential peak slope at ∼4.32 V, possibly arising from a thin surface coating of sparse LiOH or other impurities on the commercial Li₂O₂ particles, must first be overcome[49]. After that, the remaining part of the charge of Li₂O₂ is performed at the high potential of ∼3.95 V. In the G4-UH₂O₂ electrolyte; however, the charge barrier is circumvented well and the charge potential shifts to the low value of ∼3.48 V. After full charge, there is no Li₂O₂ remaining in the cathode, evidenced from the disappearance of the Li₂O₂ diffraction peaks

(Supplementary Fig. 10) and the disappearance of the Li₂O₂ particles on the cathode as observed by SEM (Supplementary Fig. 11). Accordingly, the important role of UH₂O₂ in assisting the preloaded Li₂O₂ decomposition is verified.

## Discussion

For an H₂O-containing Li-O₂ battery based on LiOH compounds as products, we developed an accessible route of introducing

$H_2O_2$ solution in the electrolyte to realize a large reduction of charge potential (from $\sim$4.16 V to $\sim$3.50 V). The crucial role of $H_2O_2$ was confirmed to greatly assist the decomposition of either solid LiOH compounds at the cathode or liquid LiOH in the electrolyte. Certain factors may be implicated in improving the charge ability in the aqueous Li-$O_2$ battery. Most importantly, for a non-aqueous Li-$O_2$ battery with $Li_2O_2$ products, we first proposed an organic $H_2O_2$ compound (urea hydrogen peroxide) without any $H_2O$ as the electrolyte additive to significantly decrease the charge potential (to $\sim$3.26 V, among the best reported) and moreover to circumvent the problem of undesired $H_2O$ corrosion of the Li metal anode. Avoiding the damage to the Li metal anode ensured the enhancement of the cycling performance and the safety of the Li-$O_2$ battery. With the aid of the novel additive, the preloaded $Li_2O_2$ becomes easier to decompose, which aids the potential development of Li metal-free Li ion $O_2$ batteries. It should be emphasized that high performance is realized at the carbon-based cathode without any additional catalysts. Although much additional work should be conducted to understand the detailed mechanisms, these results are believed to contribute to the development of pure carbon-based cathodes for Li-$O_2$ batteries, enabling the benefits of superior electron conductivity, high energy density and low cost, and should trigger further efforts to exploring other possible peroxides in addition to pointing to broader design principles for electrolytes.

## Methods

**Cathode preparation.** KB carbon was used as the cathode material. Polytetrafluoroethylene (PTFE) was used as the binder. The weight ratio between KB and PTFE was 85:15. The cathodes were prepared by rolling the KB and PTFE paste in ethanol and then pressing the paste onto hydrophobic carbon papers (diameter of 7 mm). The mass loading of KB was $\sim$1.0 mg cm$^{-2}$. For the preparation of LiOH-preloaded cathodes, commercial LiOH particles were first ball-milled in an Ar atmosphere and then mixed with KB to obtain the LiOH-preloaded KB. The weight ratio between LiOH and KB was 1:1. For the preparation of $Li_2O_2$-preloaded cathodes, commercial $Li_2O_2$ particles and KB were mixed through ball-milling in an Ar atmosphere. The weight ratio between $Li_2O_2$ and KB was 1:1. Following a method similar to the KB cathode preparation, the LiOH-preloaded KB cathodes and the $Li_2O_2$-preloaded KB cathodes were completed. Before battery assembly, all the cathodes were dried in a vacuum oven at 80 °C for 12 h.

**Electrolyte.** Tetraglyme (G4) was used as the electrolyte solvent. Before utilization, the G4 was dried by molecular sieves for 7 days. After drying in a vacuum over at 120 °C for 24 h, LiTFSI was used as the lithium salt. The molar concentration of the G4 electrolyte was 1 M. The G4-$H_2O$ or G4-$H_2O_2$ electrolyte was prepared by adding $H_2O$ or $H_2O_2$ aqueous solution (30 wt%) into the G4 electrolyte. The weight percentage of the $H_2O$ or $H_2O_2$ aqueous solution in each electrolyte was 30 wt%. For the preparation of the G4-U$H_2O_2$ electrolyte, the urea hydrogen peroxide was dissolved in the dried G4 electrolyte at a weight percentage of 5 wt%.

**Cell assembly.** The Li-$O_2$ pouch cell with LiSICON film to prohibit $H_2O$ crossover towards the Li metal anode was fabricated following the process described in previous work[43]. The coin-type Li-$O_2$ cell was formed in an Ar-filled glove box ($<$1.0 p.p.m. of $H_2O$ and 1.0 p.p.m. of $O_2$) using a 2032 coin cell with 6 holes on the top. A glassy fibre filter paper was adopted as the separator. The Li metal anode was the anode. The electrolyte volume was 30 μl for the LiSICON-based pouch cell and 50 μl for the coin cell. The assembled cells were stored in sealed glass chambers with volume capacities of 650 ml. Before electrochemical tests, the chambers were purged with $O_2$ (99.999%) for at least 2 h. The relative humidity was controlled at $\sim$75% by a saturated NaCl solution.

**Characterization and measurements.** The galvanostatic discharge/charge was conducted on a Hokuto discharging/charging system at 25 °C. The specific capacities and current densities are calculated based on the mass of KB in the cathodes. The *in situ* DEMS system used a custom-built cell connected to equipment from Perkin-Elmer (Clarus 680 and SQ 8S)[50]. The electrochemical impedance spectroscopy was performed on a Solartron (SI 1260) workstation at 25 °C. The frequency range was $10^6$–$10^{-2}$ Hz. The amplitude was $\pm$5 mV. For *ex situ* XRD measurements, SEM measurement and the titration experiment of the discharged/recharged cathodes, the cells were disassembled in an Ar-filled glove box and the cathodes were extracted. The cathodes were further thoroughly washed by dimethyl ether three times to remove residual solvent and lithium salt and finally dried in a vacuum chamber connected to the glove box. XRD measurements were performed on a Bruker D8 Advance diffractometer by sealing the cathodes with a Kapton polyimide film. SEM images were taken using a JSM-6700F instrument. The titration experiments were conducted following the method of McCloskey et al.[44] FTIR measurements were performed on an FT/IR-6200 spectrometer (JASCO Corp.).

**Data availability.** The authors declare that all the relevant data are available within the paper and its Supplementary Information file or from the corresponding author upon reasonable request.

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

## Acknowledgements

The financial support from the ALCA of Japan project, the National Basic Research Program of China (2014CB932300) and the NSF of China (21373111) are acknowledged.

## Author contributions

S.W. designed and performed the experiments. Y.Q. conducted the SEM and FTIR studies. S.Y. performed the *in situ* DEMS test. All the authors contributed to the discussion. S.W. wrote the manuscript and H.Z. supervised the work.

## Additional information

**Competing interests:** The authors declare no competing financial interests.

**Publisher's note**: 

