## [Peer Review File · Nature Communications]

Reviewers' comments:

Reviewer #1 (Remarks to the Author):

This is an interesting as well as important work in the field of Li-O₂ battery and the paper is well written. The authors demonstrate that H₂O₂ plays a significant role in the large reduction of charge potential for decomposing LiOH in Li-O₂ battery without using any other metal catalysts in carbon-based cathode. This is the first report and it does provide insights into the understanding of mechanism on H₂O influence for Li-O₂ battery. More interestingly, they find an organic chemical (urea hydrogen peroxide) as electrolyte additive in Li-O₂ battery to reduce the charge potential by virtue of H₂O₂ effect. In this case, H₂O generally existing in H₂O₂ solution is avoided not to contaminate Li metal anode. Ultralow charge potential of 3.26 V as well as good cycling performance is obtained. This work shows enough novelty and very enlightening significance.

The following points are suggested to improve the article.

1. In Figure 2, the authors presented reduced charge potential for decomposing prefilled LiOH in cathode and dissolved LiOH in electrolyte in the presence of H₂O₂. The decomposition of the prefilled LiOH in cathode was evidenced by XRD measurement. However, they did not provide data to confirm whether the dissolved LiOH is totally decomposed after charge.
2. SEM analysis should be conducted to confirm the decomposition of prefilled Li₂O₂ at low charge potential (Figure S4). And in the reviewer's opinion, the improvement of UH₂O₂ additive for decomposing the prefilled commercial Li₂O₂ is important and the results should be moved to the main text, not in the Supplementary Information.
3. In Figure 5c, what do the grey sphere corresponding to?
4. What about the ion conductivity of the electrolyte with UH₂O₂ additive? Is the ion conductivity changed after UH₂O₂ addition?
5. In Figure 1a and Figure S4, what is the current density based on the weight of KB in cathode? This value should be clearly given.

Reviewer #2 (Remarks to the Author):

The authors report on the deleterious effects of the formation of LiOH in water containing electrolytes in Li-O₂ batteries and suggest H₂O₂-like oxidants as a way to remove LiOH. They show that urea-H₂O₂ as a way to improve the charge features. The results are impressive; however lacks a key missing piece to complete the story — this is quantitative measurements of pressure rise/decay, DEMS, titration of Li₂O₂ as a way to quantify the actual discharge and charge electrochemistry. Without this information, the manuscript cannot be considered for publication.

1) The abstract is highly misleading. The authors actually are looking at the formation of parasitic LiOH-type compounds at the anode/cathodes in electrolytes containing water. It gives the reader a false impression that they are potentially considering LiOH as a discharge product. This should be modified in the abstract. Also, in the first line, LiOH should be removed as this once again gives the impression that LiOH is a potential discharge product.

2) Noble metals and oxides (Pt, Au, Pd, Ru and RuO₂ etc.), 22-27 transition metals and compounds (MnO₂, TiC, Ti₄O₇, 54 Cu₂O, FeOOH, NiOOH, Ni₂CoO₄ etc.) were introduced to construct the carbon-based composite cathodes, respectively. — It is important to remember that catalysts should be such that they do not break O-O bond. See ACS Energy Lett. 1, 162. Suitable weak bonding characteristics are required to selectively activate 2e⁻ Li₂O₂ vs 4e⁻ Li₂O.

The smileys in Figure 3 and 4 are too gimmicky. The authors should remove this and use an X and tick instead to denote the effects.

3) "Recently, Li et al. in our group demonstrated a novel route to achieve the ultralow charge potential of ~3.2 V by introducing hundred ppm of H₂O in the dimethyl sulfoxide (DMSO)-based

electrolyte and constructing a composite cathode (MnO₂ and Ru particles supported Super P carbon).” It is worth highlighting that several groups (ref. 7, Gasteiger group) demonstrated the role of water in enhancing capacity, however, they suffered from larger overpotentials and this was addressed by Li et al.

4) How confident are the authors of the reaction mechanism of the LiOH mediated oxidation? In light of all the issues relating to the LiOH chemistry, it is worthwhile for the authors to remind the readers that the chemistry is still 2e⁻ reduction of oxidation in the work of Li et al. Perhaps, even downplay the LiOH.

5) Page 4 — challenged the reversibility of Li-O₂ battery chemistry — add “with LiOH as the discharge product”

6) Figure 1 — corresponding pressure rise/decay measurements need to be done to extract e⁻/O₂. Without quantitative measurements, it is nearly impossible to conclude anything about the parasitic electrochemistry and chemistry.

7) The results of Figure 5 are impressive, but authors must quantify the pressure rise/decay to quantify e⁻/O₂. See the work from McCloskey et al. This is a crucial aspect that needs to be addressed before these results could be used to conclude enhanced rechargeability.

Reviewer #3 (Remarks to the Author):

This manuscript claims a lowering of the charge overpotential of a lithium-oxygen battery by adding H₂O₂ to the electrolyte. Particularly, the authors state that H₂O₂ can reversibly decompose LiOH in a lithium-oxygen battery (either LiOH formed from letting the discharge product Li₂O₂ react with ambient water, or LiOH prepacked in the cathode), and that an anhydrous urea-H₂O₂ enables reversible Li₂O₂ formation/decomposition at this low charge overpotential without the deleterious effects of H₂O. The use of H₂O₂ as an electrolyte additive is novel, to my knowledge. The supporting information and methods descriptions are sufficient to reproduce the work, other than electrolyte volume, as mentioned below.

The authors clearly show that electrolytes containing H₂O₂ and urea-H₂O₂ see lower overpotentials on charge, however, they do not provide sufficient evidence to ascribe this lower charge overpotential to the same reversible lithium-oxygen electrochemistry that occurs without their additives. Their evidence that a reversible Li-O₂ electrochemistry is occurring is spectroscopic signatures (or the lack thereof) of the discharge products before and after charging. A lack of a signal for LiOH on a cathode after charging, for example, could also be ascribed to a parasitic reaction having formed a soluble species. In determining reversibility of a lithium-oxygen battery, it is important to quantitatively characterize the consumption and evolution of oxygen and the formation and decomposition of the discharge product, Li₂O₂. Without quantitative gas analysis and titrations for the discharge products proving that a 2e⁻/O₂ process is maintained on both discharge and charge, a lowering of the charge overpotential and a disappearance of spectroscopic signals could similarly be ascribed to new irreversible side reactions, rather than the desired reversible Li-O₂ electrochemistry. Of note, showing multiple cycles (50 cycles) with similar performance could simply point toward steady decomposition in a large electrolyte volume (electrolyte volume was not given), rather than steady high performance. Thus, I believe this article needs additional evidence to support the claim that the observed low charge overpotential indeed corresponds to a reversible lithium-oxygen electrochemistry.

Additionally, the article does not give a proposed mechanism for the performance-enhancing effect of H₂O₂ in a lithium-oxygen battery. Given that H₂O₂ is already present when Li₂O₂ is reacted with ambient water to form LiOH, I am skeptical as to the proposed benefit of additional H₂O₂, as well as its stability, and thus would appreciate a proposed method of action of the H₂O₂. I also believe a proposed mechanism, whether it is a discussion of a catalytic pathway or solubility enhancements, etc., is critical for this article to influence a wider field, as it could point to broader design principles for

electrolytes in electrochemical systems.

For these reasons, I do not believe this manuscript should be published in its current form.

Reviewers' comments:

Reviewer #1 (Remarks to the Author):

Authors have addressed all the comments. I would support for the publication of this work, which brings innovation thinking to the current Li O₂ battery research.

Reviewer #2 (Remarks to the Author):

The authors have done an excellent job of addressing the reviewer concerns including the new experiments that were carried out to address the concerns regarding yield and rechargeability. In light of this, I think the manuscript is now suitable for publication.

Reviewer #3 (Remarks to the Author):

The authors have made significant helpful revisions to their paper, but I believe there are still two points from the original reviews that have not been adequately addressed:

1) Confusion as to when LiOH is the discharge product being oxidized on charge and when Li₂O₂ is the discharge product being oxidized on charge.

My understanding is that the primary goal of this paper, according to the abstract and the concluding paragraph, is to report if adding hydrogen peroxide (and in particular an anhydrous urea-hydrogen peroxide) to a lithium-oxygen battery can enable the reversible oxidation of lithium hydroxide. This work is seen in Figures 1-4. According to the introduction, it seems this goal is an extension of the authors' group's recent work on the promotion of LiOH as the primary discharge product by using MnO₂ and Ru in their cathode and including H₂O in their electrolyte. In addition, the authors present whether an anhydrous urea-hydrogen peroxide can serve as a helpful additive in oxidizing Li₂O₂, as seen in Figures 5-7. I think the authors still need to be clearer in both the abstract and the text (particularly introduction and conclusion) that they are discussing at times a LiOH-based lithium-oxygen electrochemistry and at times a Li₂O₂-based electrochemistry. I think both ideas are interesting and important to present; I just believe the authors should more clearly separate the two, especially as they only have evidence of a reversible, oxygen-forming oxidation reaction when Li₂O₂ is the discharge product being oxidized on charge.

2) The critical DEMS results to support reversible LiOH oxidation are still missing, and trying to approximate DEMS results with titration results is incorrect.

In our first reviews, another reviewer and I asked that differential electrochemical mass spectrometry (DEMS) be employed to verify that on charge H₂O₂ is indeed reversibly oxidizing LiOH to oxygen. Here, the authors provide titration results of extracted cathodes to show that LiOH, present before charge, is absent after charge. While this additional data is useful, the conclusion that this data can serve as a proxy for DEMS is incorrect.

Titration results and DEMS results provide two different pieces of information – the formation and decomposition of Li₂O₂ and the formation and decomposition of oxygen, respectively. For example, if more Li₂O₂ has been consumed after charge (as measured via titration) than O₂ has been evolved (as measured via DEMS), this implies that some Li₂O₂ has been oxidized to parasitic products. This idea was recently presented in studying the oxidation of LiOH in Burke et al., ACS Energy Letters, 2016, 1,

747-756. In that article, the authors looked at the reversibility of I⁻ and H₂O-induced LiOH formation in lithium-oxygen batteries. Figure 3(c-d) show that a battery with a certain electrolyte exhibits the decomposition of nearly all LiOH on charge. While this seems to imply reversibility, Figure 2(g-i) clearly show that no oxygen is evolved when a battery with that certain electrolyte is charged. Thus, Burke et al. shows that the absence of LiOH after charge is not sufficient evidence to claim reversible oxidation of LiOH.

In the manuscript at hand, I still feel the authors have failed to corroborate the statement in the abstract that they, "find that the hydrogen peroxide (H₂O₂) aqueous solution added in the electrolyte can effectively promote the decomposition of LiOH compounds at the ultralow charge potential on the catalyst-free Ketjen Black-based cathode," as the authors have indeed proven that LiOH decomposes, but have not yet proven that LiOH decomposes reversibly to oxygen, as I believe is implied.

In the rebuttal letter, the authors expressed the difficulty in measuring oxygen evolution from a cell containing hydrated H₂O₂, as there is a rapid reaction between water and the lithium metal anode that produces hydrogen. I understand this difficulty, and recommend that if the authors cannot perform accurate DEMS to check for O₂ evolution from LiOH during charge, that the authors note that while LiOH has been decomposed on charge, it is unclear if it has been reversibly oxidized to oxygen. I believe this article is still an important addition to the field with this acknowledgment.

The DEMS results presented in Figure 6 do indeed show reasonable reversibility of Li₂O₂ in a cell containing urea-H₂O₂, and I appreciate this addition.

REVIEWERS' COMMENTS:

Reviewer #3 (Remarks to the Author):

The authors have made the requested revisions, delineating between the LiOH-based electrochemistry and the Li₂O₂-based electrochemistry and adding the DEMS work for the LiOH-based electrochemistry. I agree that the manuscript is now suitable for publication.

Response to Reviewers:

Reviewer #1:..... 1
Reviewer #2:..... 8
Reviewer #3:..... 18

Reviewer #1:

This is an interesting as well as important work in the field of Li-O₂ battery and the paper is well written. The authors demonstrate that H₂O₂ plays a significant role in the large reduction of charge potential for decomposing LiOH in Li-O₂ battery without using any other metal catalysts in carbon-based cathode. This is the first report and it does provide insights into the understanding of mechanism on H₂O influence for Li-O₂ battery. More interestingly, they find an organic chemical (urea hydrogen peroxide) as electrolyte additive in Li-O₂ battery to reduce the charge potential by virtue of H₂O₂ effect. In this case, H₂O generally existing in H₂O₂ solution is avoided not to contaminate Li metal anode. Ultralow charge potential of 3.26 V as well as good cycling performance is obtained. This work shows enough novelty and very enlightening significance.

The following points are suggested to improve the article.

1. In Figure 2, the authors presented reduced charge potential for decomposing prefilled LiOH in cathode and dissolved LiOH in electrolyte in the presence of H₂O₂. The decomposition of the prefilled LiOH in cathode was evidenced by XRD measurement. However, they did not provide data to confirm whether the dissolved LiOH is totally decomposed after charge.

As the reviewer suggested, we conducted Fourier Transform Infrared Spectroscopy (FTIR) to confirm whether the dissolved LiOH could be totally decomposed after

charge. The related data is presented in Supplementary Fig. S3 and corresponding revision has been made in manuscript (Page 8).

After charge at high potential in G4-H₂O-LiOH_(l) electrolyte (Fig. 2b), the peak of LiOH at ~3680 cm⁻¹ in FTIR spectra (Supplementary Fig. S3, iii) still remains, indicating the incomplete decomposition of liquid LiOH in electrolyte. In contrast, the LiOH can be fully decomposed after charge at low potential in H₂O₂-containing electrolyte (Fig. 2b), evidenced from the disappeared peak (Supplementary Fig. S3, iv).

Supplementary Fig. 3 | FTIR spectra of (i) pristine G4-H₂O-LiOH_(l) electrolyte, (ii) G4-H₂O-LiOH_(l)-H₂O₂ electrolyte, (iii) G4-H₂O-LiOH_(l) electrolyte after charge and (iv) G4-H₂O-LiOH_(l)-H₂O₂ electrolyte after charge, respectively. The peak at ~3680 cm⁻¹ corresponds to liquid LiOH in electrolyte.

2. SEM analysis should be conducted to confirm the decomposition of prefilled Li₂O₂ at low charge potential (Figure S4). And in the reviewer's opinion, the improvement of UH₂O₂ additive for decomposing the prefilled commercial Li₂O₂ is important and the results should be moved to the main text, not in the Supplementary Information.

The SEM analysis and the related description have been added to confirm the decomposition of prefilled Li_2O_2 at low charge potential in G4- UH_2O_2 electrolyte (Supplementary Fig. S11). Considering the length of the article, it may be better to keep this part in Supplementary Information.

The corresponding revisions in Page 18 in manuscript and in Page 12-13 in Supplementary Information have been made.

Supplementary Fig. 11 | SEM images of (a) the commercial Li_2O_2 particles, (b) the Li_2O_2 -preloaded KB cathode and the charged Li_2O_2 -preloaded KB cathodes in (c) G4-based and (d) G4- UH_2O_2 electrolytes, respectively.

The diameters of commercial Li_2O_2 particles are about 0.5 – 1 μm in Supplementary Fig. S11a. After preloading Li_2O_2 on KB, the mixture of Li_2O_2 and KB can be identified from SEM image in Supplementary Fig. S11b. After full charge in G4-based and G4- UH_2O_2 electrolytes, respectively, no Li_2O_2 particles can be observed in Supplementary Fig. S11c and d, leaving voids

surrounded by KB particles. Combining the disappearance of Li_2O_2 diffraction peaks in XRD patterns (Supplementary Fig. S10), it can be confirmed that the preloaded Li_2O_2 can be completely decomposed either in G4-based electrolyte or in the G4- UH_2O_2 electrolyte.

3. In Figure 5c, what do the grey sphere corresponding to?

The grey sphere in previous Figure 5c (Figure 7c at present) corresponds to the variations of specific capacity, terminal charge and discharge voltages in the Li- O_2 cells with G4-based electrolyte during cycling. In order to distinguish the figure, we added the illustration in Figure 7c.

Figure 7 | Cycling performance of Li- O_2 cell. The selected discharge and charge profiles in (a) the G4- UH_2O_2 and (b) G4-based electrolyte, respectively. (c) The corresponding discharge and charge terminal voltages along with capacities upon cycles. Current density: $500 \text{ mA g}^{-1}_{\text{KB}}$.

4. What about the ion conductivity of the electrolyte with UH_2O_2 additive? Is the ion conductivity changed after UH_2O_2 addition?

In order to evaluate the ion conductivity of the electrolyte with UH_2O_2 additive, we carried out the electrochemical impedance spectra. The related data is shown in Supplementary Fig. S6. From the Nyquist plots, the ion conductivity σ can be calculated via the equation $\sigma = d/(R_b S)$, where d is the thickness of the applied separator (20 μm), R_b is the resistance obtained from the Nyquist plot, and S is the surface area of the stainless steel (1.767 cm^2). The calculated ion conductivities are $2.0 \times 10^{-3} \text{ S cm}^{-1}$ for G4-based electrolyte and $1.8 \times 10^{-3} \text{ S cm}^{-1}$ for G4- UH_2O_2 electrolyte, respectively. These values are comparable to the level in commercial electrolyte for Li ion batteries. After introduction of UH_2O_2 into electrolyte, the Li^+ ion conductivity is still high and there is nearly no change of ion conductivity compared with G4-based electrolyte.

Supplementary Fig. 6 | Nyquist plots of the G4-based electrolyte and G4-UH₂O₂ electrolyte with the cell structure of stainless steel (SS)/electrolyte/SS. Insets are the amplified figures.

Corresponding revision has been made in Page 13 in manuscript.

5. In Figure 1a and Figure S4, what is the current density based on the weight of KB in cathode? This value should be clearly given.

As the reviewer indicated, we have given the current density based on the weight of KB in cathode in Figure 1a and Figure S4 (Figure S8 at present). In Figure 1a and

Figure S4, the current densities are 100 mA g^{-1} based on the weight of KB. In other figures (Figure 2 and Figure 3), we also added the corresponding current densities.

Reviewer #2:

The authors report on the deleterious effects of the formation of LiOH in water containing electrolytes in Li-O₂ batteries and suggest H₂O₂-like oxidants as a way to remove LiOH. They show that urea-H₂O₂ as a way to improve the charge features. The results are impressive; however, lacks a key missing piece to complete the story — this is quantitative measurements of pressure rise/decay, DEMS, titration of Li₂O₂ as a way to quantify the actual discharge and charge electrochemistry. Without this information, the manuscript cannot be considered for publication.

Thank you for the helpful suggestions. It is important for further improvement of our work. As the reviewer's suggestions, we have conducted the quantitative measurements of in situ DEMS and titration of Li₂O₂. Also, in the manuscript, corresponding revisions have been made.

1) The abstract is highly misleading. The authors actually are looking at the formation of parasitic LiOH-type compounds at the anode/cathodes in electrolytes containing water. It gives the reader a false impression that they are potentially considering LiOH as a discharge product. This should be modified in the abstract. Also, in the first line, LiOH should be removed as this once again gives the impression that LiOH is a potential discharge product.

LiOH in the first line has been deleted and the abstract has been revised to avoid the misleading information. In some reported work related to Li-O₂ battery with H₂O/moisture in the system (*J. Am. Chem. Soc.* **2012**, *134*, 2902; *Phys. Chem. Chem. Phys.* **2016**, *18*, 24944; *Nat. Commun.* **2015**, *6*, 7843; *Science* **2015**, *350*, 530, etc.), LiOH compounds were detected to be the constituent of discharge products either as the parasitic products or as the main products. It is of the essence to address the difficult decomposition of LiOH for improving the performance of Li-O₂

battery. In our work, we find the H_2O_2 added in the electrolyte can effectively promote the decomposition of LiOH compounds at the ultralow charge potential on the catalyst-free Ketjen Black-based cathode. These encouraging results are believed to contribute to the development of high-performance Li- O_2 battery.

2) Noble metals and oxides (Pt, Au, Pd, Ru and RuO_2 etc.),²²⁻²⁷ transition metals and compounds (MnO_2 , TiC, Ti_4O_7 , Cu_2O , FeOOH, NiOOH, Ni_2CoO_4 etc.) were introduced to construct the carbon-based composite cathodes, respectively. — It is important to remember that catalysts should be such that they do not break O-O bond. See ACS Energy Lett. 1, 162. Suitable weak bonding characteristics are required to selectively activate $2\text{e}^- \text{Li}_2\text{O}_2$ vs $4\text{e}^- \text{Li}_2\text{O}$.

Thanks for reminding. Although the introduction of catalyst in cathode was reported to facilitate the reduction of charge potential, the underlying mechanism is still unclear. Also, in some cases, as reported in ACS Energy Lett. 1, 162, the employment of inappropriate catalysts that strongly bind O_2 or the discharge intermediate LiO_2 may result in the undesired shift from Li_2O_2 to Li_2O as discharge products and further lead to the poor reversibility of Li- O_2 battery. On the other hand, the heavy molecular weight and expensive price of general catalysts is inevitably subjected to the significantly decreased energy density and increased cost. And, the over strong catalysis activity of some catalysts may lead to the parasitic reactions of the electrolyte. Therefore, when designing the composite cathodes, suitable catalysts should be carefully selected. This is the reasons that we aim to improve the performance of Li- O_2 battery with catalyst-free Ketjen Black carbon-based cathode by utilizing H_2O_2 -like additive in electrolyte.

In the corresponding part of introduction, we have added description about this information (Page 3-4).

“Noble metals and oxides (Pt, Au, Pd, Ru and RuO₂ etc.),²²⁻²⁷ transition metals and compounds (MnO₂, TiC, Ti₄O₇, Cu₂O, FeOOH, NiOOH, Ni₂CoO₄ etc.) were tentatively introduced to construct the carbon-based composite cathodes, respectively.²⁸⁻³⁰ In this way, the ultimate charge potentials of ~3.5 V can be expected, but it is hard to break through this limit. Also of note, the employment of inappropriate catalysts that strongly bind O₂ or the discharge intermediate LiO₂ may result in the undesired shift from Li₂O₂ to Li₂O as discharge products and further lead to the poor reversibility of Li-O₂ battery.^{31, 32”}

3) The smileys in Figure 3 and 4 are too gimmicky. The authors should remove this and use an X and tick instead to denote the effects.

The smileys in Figure 3 and 4 have been substituted with an X and tick. The figures have been revised in manuscript.

4) “Recently, Li et al. in our group demonstrated a novel route to achieve the ultralow charge potential of ~3.2 V by introducing hundred ppm of H₂O in the dimethyl sulfoxide (DMSO)-based electrolyte and constructing a composite cathode (MnO₂ and Ru particles supported Super P carbon).” It is worth highlighting that several groups (ref. 7, Gasteiger group) demonstrated the role of water in enhancing capacity, however, they suffered from larger overpotentials and this was addressed by Li et al.

The investigation about water influence for Li-O₂ battery is of the essence and importance to realize practical Li-air batteries. Recently, related research has attracted much attention. The large enhancement of discharge capacity in the presence of water in electrolyte was mainly demonstrated by Gasteiger group and ref. 7.

As the reviewer suggested, we highlighted these valuable results and made corresponding revisions in the manuscript (Page 4).

“Recently, H₂O in Li-O₂ battery system was reported to largely enhance the discharge capacity,^{7, 33, 34} but the charge potentials were still high, generally above 4.2 V in Li-O₂ battery with a carbon cathode. Li et al. in our group demonstrated a novel route to achieve the ultralow charge potential of ~3.2 V in the dimethyl sulfoxide (DMSO)-based electrolyte containing hundred ppm of H₂O by constructing a composite cathode (MnO₂ and Ru particles supported Super P carbon).³⁵”

5) How confident are the authors of the reaction mechanism of the LiOH mediated oxidation? In light of all the issues relating to the LiOH chemistry, it is worthwhile for the authors to remind the readers that the chemistry is still 2e⁻ reduction of oxidation in the work of Li et al. Perhaps, even downplay the LiOH.

In the presence of excessive H₂O in Li-O₂ battery system, Aetukuri et al. (*Nature Chem.* 2015, **7**, 50) and Gasteiger group (*Electrochem. Soc.* 2015, **162**, A573), respectively, reported the discharge product was mainly Li₂O₂ and no LiOH on cathode was detected. When there was MnO₂ in cathode (*Nat. Commun.* 2015, **6**, 7843 in our group), the main discharge product was converted to LiOH. This result was explained by the broken reaction equilibrium ($\text{Li}_2\text{O}_2 + 2\text{H}_2\text{O} \rightleftharpoons 2\text{LiOH} + \text{H}_2\text{O}_2$) after the decomposition of H₂O₂ over MnO₂. While enough H₂O/moisture was introduced (*Chem. Commun.* 2015, **51**, 16860; *Adv. Funct. Mater.* 2016, **26**, 3291), LiOH compounds were identified to be the only discharge products on MnO₂-containing cathode. After the charge process, the LiOH diffraction peak in XRD patterns, FTIR analysis and their particles in SEM image disappeared and O₂ evolved (GC-MS). Based on these results, it is believed that the reaction mechanism of the LiOH

mediated oxidation predominates in charge process for these Li-O₂ battery with Ru/MnO₂-containing cathode and water-containing electrolyte.

In the work of Li et al., the formation of LiOH was proposed through an electrochemical reaction and a chemical reaction. The electrochemical reaction corresponded to the generation of Li₂O₂ from O₂ reduction. The chemical reaction corresponded to the reaction between Li₂O₂ and H₂O (Li₂O₂+2H₂O \rightleftharpoons 2LiOH+H₂O₂; 2H₂O₂ \rightarrow 2H₂O+O₂ (catalyzed by MnO₂ in cathode)). The value of e⁻/O₂ during discharge process was calculated to be about 2 through titration measurements. This should be rational because only an electrochemical reduction of O₂ to Li₂O₂ occurred during discharge process.

During charge process corresponding to the LiOH oxidation, Li et al. did not study the e⁻/O₂ value. In my opinion, in their proposed Li-O₂ battery system, the oxidative decomposition of LiOH was realized over the catalysis of the efficient Ru nanoparticles and thus the electrochemistry may not be 2e⁻ process. Additional work should be performed to figure out the detailed mechanism.

As the reviewer suggested, we added the description about the 2e⁻ discharge process to remind the readers (Page 4).

“The primary electrochemical reactions (the formation of Li₂O₂) during discharge were converted to the formation of lithium hydroxide (LiOH) via the catalysis effect of MnO₂ in cathode towards the reactions between Li₂O₂ and H₂O. **The discharge process was proposed to involve a 2e⁻ electrochemical reaction and a following chemical reaction.** During charge, Ru in cathode decomposes the LiOH at the low charge potential.”

6) Page 4 — challenged the reversibility of Li-O₂ battery chemistry — add “with LiOH as the discharge product”

The words have been added in order to accurately show the meaning.

7) Figure 1 — corresponding pressure rise/decay measurements need to be done to extract e^-/O_2 . Without quantitative measurements, it is nearly impossible to conclude anything about the parasitic electrochemistry and chemistry.

In order to quantitatively examine the charge process corresponding to the LiOH oxidation in the H_2O_2 -containing electrolyte in Fig. 1, we conducted the titration experiment to determine the consumed amount of LiOH in varied charge states. The titration experiment was employed instead of the pressure rise/decay measurements because there was large amount of H_2O in the electrolyte (20-30%) and the Li metal anode in the custom-built Li- O_2 cell for pressure rise/decay measurement was initially designed without any protection and inevitably would suffer from severe corrosion by the H_2O . Based on the work of McCloskey on investigation of titration and in situ DEMS measurement (J. Phys. Chem. Lett. 2013, 4, 2989), it is reasonable to indicate quantitative results of in situ DEMS measurement according to the titration results. Fig. 1b showed the titration results.

The KB-based cathodes in situ loading solid LiOH compounds were obtained by firstly discharging Li- O_2 pouch cells in the dry electrolyte to 1.50 mAh (Fig. 1a i) with Li_2O_2 as the products evidenced from the X-ray diffraction (XRD) pattern in Fig. 1c, extracting the discharged cathodes in an Ar glove box and leaving the cathodes in an Ar atmosphere with a relative humidity of 75% for 7 days. The XRD pattern in Fig. 1d confirms all the Li_2O_2 on the discharged cathode is converted to the mixture of LiOH and $LiOH \cdot H_2O$. The amount of these LiOH compounds (n_{LiOHii}) was titrated following the method of McCloskey. Another new Li- O_2 pouch cells (Fig. 1a ii) were assembled with these cathodes and H_2O_2 -containing electrolytes. When the cells were charged to certain capacities (0.2, 0.5, 0.75, 0.85, 1.22 and 1.356 mAh, respectively), the

residual amount of LiOH compounds was titrated and the consumed amount of LiOH compounds could be obtained by subtracting the LiOH compounds at each point from $n_{\text{LiOH}ii}$.

As shown in Fig. 1b, at each point during charge process, the calculated amount of LiOH compounds consumed is almost equal to the theoretical value and the relationship of consumed amount and the charge capacity is linearly dependent. These quantitative results indicate the oxidation of LiOH compounds dominates in the charge process and there is no obvious parasitic electrochemistry and chemistry.

Corresponding revisions have been made in Page 6-7 in manuscript.

Figure 1 | The reduced charge potentials for decomposing solid LiOH compounds on KB-based cathode in the G4-based electrolyte by the action of H_2O_2 . (a) The discharge and charge profiles at $100 \text{ mA g}^{-1}_{\text{KB}}$. During charging, solid LiOH compounds can be oxidatively decomposed at a low potential of 3.50 V (corresponding to the overpotential of

0.54 V) in the presence of H_2O_2 solution in the electrolyte. This is much lower than the value (1.20 V) without H_2O_2 . (b) The quantify of LiOH compounds consumption, n_{LiOH} , c , during charge determined by titration measurement. (c-e) XRD patterns of cathodes corresponding to the different state (i, ii, iii and iv) in Figure 1a. The Li-O₂ pouch cell is firstly discharged to 1.5 mAh (corresponding to $\sim 4000 \text{ mAh g}^{-1}_{\text{KB}}$ and 3.5 mAh cm^{-2}) in the dry electrolyte to produce Li_2O_2 in cathode (c). The Li_2O_2 is converted to the mixture of LiOH and $\text{LiOH}\cdot\text{H}_2\text{O}$ (d) by keeping the cathode (i) in the Ar atmosphere with a relative humidity of 75% for 7 days. After charging, the LiOH and $\text{LiOH}\cdot\text{H}_2\text{O}$ are reversibly oxidized at low charge potential in the presence of H_2O_2 in the electrolyte, evidenced from the disappearance of their diffraction peaks in (e) iv. Whereas, $\text{LiOH}\cdot\text{H}_2\text{O}$ remains undecomposed in the absence of H_2O_2 in the electrolyte (e) iii. Current density: $100 \text{ mA g}^{-1}_{\text{KB}}$. LISICON film preventing H_2O penetration was employed to fabricate the Li-O₂ pouch cells.

8) The results of Figure 5 are impressive, but authors must quantify the pressure rise/decay to quantify e^-/O_2 . See the work from McCloskey et al. This is a crucial aspect that needs to be addressed before these results could be used to conclude enhanced rechargeability.

As the reviewer suggested, according to the work from McCloskey et al. (J. Phys. Chem. Lett. 2013, 4, 2989), we carried out the in situ DEMS measurement to observe the pressure rise/decay and titration experiment to quantify e^- involved in the reaction. Because the preliminary custom-built cell for DEMS measurement in the lab has too large space volume for gas storage ($\sim 10 \text{ mL}$, nearly 80 times than that of McCloskey in J. Phys. Chem. Lett. 2011, 2, 1161 (125 μL)) and the defective design makes that the evolved gas during charge is unable to be completely transferred to MS for quantifying gas and calculating e^-/O_2 , instead, we used titration to evaluate the $e^-/\text{Li}_2\text{O}_2$ and combined DEMS for qualitative gas analysis. The value of e^-/O_2 was confirmed to be in linear proportion to the value of $e^-/\text{Li}_2\text{O}_2$ when the overpotentials held at low values in the article of McCloskey. Accordingly, the value of e^-/O_2 can be estimated. The corresponding data was provided in Fig. 6 in manuscript. After discharging the cell to 0.1, 0.2, 0.3, 0.4 and 0.5 mAh at 0.05 mA, respectively, the

quantify of Li_2O_2 formation was determined by titration method (Fig. 6c). The yields of the formed Li_2O_2 (the amount of Li_2O_2 titrated divided by the amount of Li_2O_2 expected given the Coulometry) hold at $\sim 91\%$. This value is similar to the yield of McCloskey and indicates a little possible side reactions corresponding to electrolyte or cathode decomposition. The value of $e^-/\text{Li}_2\text{O}_2$ can be estimated to be 2.15, close to the theoretical value of 2. Upon charge, the gas evolution rate of O_2 and CO_2 are presented in Fig. 6b. It can be seen that O_2 evolution was detected from the beginning and continued along the charge potential of ~ 3.3 V during the whole charge process. No evolution of CO_2 was detected. The amounts of Li_2O_2 consumed during charge were analyzed by titration measurement (Fig. 6d). Li_2O_2 oxidation follows a $\sim 2.14e^-/\text{Li}_2\text{O}_2$ process during the whole charge process. This value approaches to the ideal value of 2 and clearly demonstrates that the significant reduction of charge overpotential due to the UH_2O_2 additive in electrolyte can prohibit the high charge potential-induced side reaction and the introduction of UH_2O_2 can improve the charge performance. The DEMS and titration results suggest that the charge reactions are dominated by evolution of O_2 and consumption of Li_2O_2 . Therefore, the high rechargeability of the Li- O_2 cell with G4- UH_2O_2 electrolyte can be concluded.

Corresponding revisions have been made in Page 14-15 in manuscript.

Figure 6 | (a) Discharge and charge profiles of Li-O₂ cell with G4-UH₂O₂ electrolyte during the in situ DEMS (differential quantitative mass spectrometry) measurement. (b) Corresponding O₂ and CO₂ evolution rate during charge. (c) The quantify of Li₂O₂ formation, $n_{\text{Li}_2\text{O}_2, \text{d}}$, during discharge determined by titration. (d) The quantify of Li₂O₂ consumption, $n_{\text{Li}_2\text{O}_2, \text{c}}$, during charge determined by titration.

Reviewer #3:

This manuscript claims a lowering of the charge overpotential of a lithium-oxygen battery by adding H_2O_2 to the electrolyte. Particularly, the authors state that H_2O_2 can reversibly decompose LiOH in a lithium-oxygen battery (either LiOH formed from letting the discharge product Li_2O_2 react with ambient water, or LiOH prepaced in the cathode), and that an anhydrous urea- H_2O_2 enables reversible Li_2O_2 formation/decomposition at this low charge overpotential without the deleterious effects of H_2O . The use of H_2O_2 as an electrolyte additive is novel, to my knowledge. The supporting information and methods descriptions are sufficient to reproduce the work, other than electrolyte volume, as mentioned below.

Thank you for the encouraging comments. Based on your suggestions, we have conducted additional experiments and made corresponding revisions in manuscript.

The authors clearly show that electrolytes containing H_2O_2 and urea- H_2O_2 see lower overpotentials on charge, however, they do not provide sufficient evidence to ascribe this lower charge overpotential to the same reversible lithium-oxygen electrochemistry that occurs without their additives. Their evidence that a reversible Li- O_2 electrochemistry is occurring is spectroscopic signatures (or the lack thereof) of the discharge products before and after charging. A lack of a signal for LiOH on a cathode after charging, for example, could also be ascribed to a parasitic reaction having formed a soluble species.

In order to confirm whether the lower charge overpotential in the presence of H_2O_2 in electrolyte is attributed to the same reversible lithium-oxygen electrochemistry that occurs without the additives, additional evidences were carried out including quantitative titration experiment to examine the charge process and FTIR analysis of

electrolyte after charge to examine whether there is soluble species from parasitic reaction.

Quantitative titration experiments (refer to the work of McCloskey et al. in J. Phys. Chem. Lett. 2013, 4, 2989) corresponding to the LiOH oxidation in the H₂O₂-containing electrolyte in Fig. 1, were conducted to determine the consumed amount of LiOH in varied charge states. Fig. 1b showed the titration results.

Figure 1 | The reduced charge potentials for decomposing solid LiOH compounds on KB-based cathode in the G4-based electrolyte by the action of H₂O₂. (a) The discharge and charge profiles at 100 mA g⁻¹_{KB}. During charging, solid LiOH compounds can be oxidatively decomposed at a low potential of 3.50 V (corresponding to the overpotential of 0.54 V) in the presence of H₂O₂ solution in the electrolyte. This is much lower than the value (1.20 V) without H₂O₂. (b) The quantify of LiOH compounds consumption, $n_{\text{LiOH},c}$, during charge determined by titration measurement. (c-e) XRD patterns of cathodes corresponding to the different state (i, ii, iii and iv) in Figure 1a. The Li-O₂ pouch cell is firstly discharged to 1.5 mAh (corresponding to ~4000 mAh g⁻¹_{KB} and 3.5 mAh cm⁻²) in the dry electrolyte to

produce Li_2O_2 in cathode (c). The Li_2O_2 is converted to the mixture of LiOH and $\text{LiOH}\cdot\text{H}_2\text{O}$ (d) by keeping the cathode (i) in the Ar atmosphere with a relative humidity of 75% for 7 days. After charging, the LiOH and $\text{LiOH}\cdot\text{H}_2\text{O}$ are reversibly oxidized at low charge potential in the presence of H_2O_2 in the electrolyte, evidenced from the disappearance of their diffraction peaks in (e) iv. Whereas, $\text{LiOH}\cdot\text{H}_2\text{O}$ remains undecomposed in the absence of H_2O_2 in the electrolyte (e) iii. Current density: $100 \text{ mA g}^{-1}_{\text{KB}}$. LiSICON film preventing H_2O penetration was employed to fabricate the Li- O_2 pouch cells.

The KB-based cathodes in situ loading solid LiOH compounds were obtained by firstly discharging Li- O_2 pouch cells in the dry electrolyte to 1.50 mAh (Fig. 1a i) with Li_2O_2 as the products evidenced from the X-ray diffraction (XRD) pattern in Fig. 1c, extracting the discharged cathodes in an Ar glove box and leaving the cathodes in an Ar atmosphere with a relative humidity of 75% for 7 days. The XRD pattern in Fig. 1d confirms all the Li_2O_2 on the discharged cathode is converted to the mixture of LiOH and $\text{LiOH}\cdot\text{H}_2\text{O}$. The amount of these LiOH compounds (n_{LiOHii}) was titrated following the method of McCloskey. Another new Li- O_2 pouch cells (Fig. 1a ii) were assembled with these cathodes and H_2O_2 -containing electrolytes. When the cells were charged to certain capacities (0.2, 0.5, 0.75, 0.85, 1.22 and 1.356 mAh, respectively), the residual amount of LiOH compounds was titrated and the consumed amount of LiOH compounds could be obtained by subtracting the LiOH compounds at each point from n_{LiOHii} .

As shown in Fig. 1b, at each point during charge process, the calculated amount of LiOH compounds consumed is almost equal to the theoretical value and the relationship of consumed amount and the charge capacity is linearly dependent. These quantitative results indicate the oxidation of LiOH compounds dominates in the charge process.

The examination of electrolyte after charge is presented in Supplementary Fig. S1. Compared with the pristine G4- H_2O - H_2O_2 electrolyte (i), there is nearly no change in

FTIR spectra for G4-H₂O-H₂O₂ electrolytes after charge (ii), indicating no obvious decomposition of electrolyte occurred and no soluble species form.

Based on these results, it can be concluded that the low charge potential in electrolyte with H₂O₂ additive corresponds to the oxidation decomposition of LiOH compounds.

Corresponding revisions have been made in Page 6-7 in manuscript.

Supplementary Fig. 1 | FTIR spectra of (i) pristine G4-H₂O-H₂O₂ electrolyte and (ii) G4-H₂O-H₂O₂ electrolyte after charge, respectively.

In determining reversibility of a lithium-oxygen battery, it is important to quantitatively characterize the consumption and evolution of oxygen and the formation and decomposition of the discharge product, Li₂O₂. Without quantitative gas analysis and titrations for the discharge products proving that a 2e⁻/O₂ process is maintained on both discharge and charge, a lowering of the charge overpotential and a

disappearance of spectroscopic signals could similarly be ascribed to new irreversible side reactions, rather than the desired reversible Li-O₂ electrochemistry.

As the reviewer suggested, according to the work from McCloskey et al. (J. Phys. Chem. Lett. 2013, 4, 2989), we carried out the in situ DEMS measurement to observe the pressure rise/decay and titration experiment to quantify e⁻ involved in the reaction. Because the preliminary custom-built cell for DEMS measurement in the lab has too large space volume for gas storage (~10 mL, nearly 80 times than that of McCloskey in J. Phys. Chem. Lett. 2011, 2, 1161 (125 μL)) and the defective design makes that the evolved gas during charge is unable to be completely transferred to MS for quantifying gas and calculating e⁻/O₂, instead, we used titration to evaluate the e⁻/Li₂O₂ and combined DEMS for qualitative gas analysis. The value of e⁻/O₂ was confirmed to be in linear proportion to the value of e⁻/Li₂O₂ when the overpotentials held at low values in the article of McCloskey. Accordingly, the value of e⁻/O₂ can be estimated. The corresponding data was provided in Fig. 6 in manuscript. After discharging the cell to 0.1, 0.2, 0.3, 0.4 and 0.5 mAh at 0.05 mA, respectively, the quantify of Li₂O₂ formation was determined by titration method (Fig. 6c). The yields of the formed Li₂O₂ (the amount of Li₂O₂ titrated divided by the amount of Li₂O₂ expected given the Coulometry) hold at ~ 91%. This value is similar to the yield of McCloskey and indicates a little possible side reactions corresponding to electrolyte or cathode decomposition. The value of e⁻/Li₂O₂ can be estimated to be 2.15, close to the theoretical value of 2. Upon charge, the gas evolution rate of O₂ and CO₂ are presented in Fig. 6b. It can be seen that O₂ evolution was detected from the beginning and continued along the charge potential of ~3.3 V during the whole charge process. No evolution of CO₂ was detected. The amounts of Li₂O₂ consumed during charge were analyzed by titration measurement (Fig. 6d). Li₂O₂ oxidation follows a ~2.14e⁻/Li₂O₂ process during the whole charge process. This value approaches to the ideal value of 2 and clearly demonstrates that the significant

reduction of charge overpotential due to the UH_2O_2 additive in electrolyte can prohibit the high charge potential-induced side reaction and the introduction of UH_2O_2 can improve the charge performance. The DEMS and titration results suggest that the charge reactions are dominated by evolution of O_2 and consumption of Li_2O_2 . Therefore, the high rechargeability of the Li-O_2 cell with G4- UH_2O_2 electrolyte can be concluded.

Corresponding revisions have been made in Page 14-15 in manuscript.

Figure 6 | (a) Discharge and charge profiles of Li-O_2 cell with G4- UH_2O_2 electrolyte during the in situ DEMS (differential quantitative mass spectrometry) measurement. (b) Corresponding O_2 and CO_2 evolution rate during charge. (c) The quantify of Li_2O_2 formation, $n_{\text{Li}_2\text{O}_2, \text{d}}$, during discharge determined by titration. (d) The quantify of Li_2O_2 consumption, $n_{\text{Li}_2\text{O}_2, \text{c}}$, during charge determined by titration.

Of note, showing multiple cycles (50 cycles) with similar performance could simply point toward steady decomposition in a large electrolyte volume (electrolyte volume

was not given), rather than steady high performance. Thus, I believe this article needs additional evidence to support the claim that the observed low charge overpotential indeed corresponds to a reversible lithium-oxygen electrochemistry.

The electrolyte volume added in the 2030-coin cell is 50 μL . This is a general value in most of the reported work and is much less than that (0.2 - 1 mL) in Science, 2015, 350, 530. The electrolyte volume has been added in the part of Cell assembly in manuscript.

With regard to the reversibility of this Li-O₂ cell with G4-UH₂O₂ electrolyte, the in situ DEMS and titration results (Fig. 6) confirmed the discharge and charge reactions corresponded to the consumption and evolution of O₂ and the formation and oxidation of Li₂O₂. Accordingly, the observed low charge overpotential corresponds to a reversible Li-O₂ electrochemistry.

The electrolytes before and after discharge/charge process were analyzed by FTIR analysis to confirm whether there was severe electrolyte decomposition. The results were presented in Supplementary Fig. S8. Compared with the pristine G4-UH₂O₂ electrolyte (Supplementary Fig. S8 iii), there is nearly no change in FTIR spectra for G4-UH₂O₂ electrolytes after discharge (iv) and charge (v), indicating no obvious decomposition of electrolyte occurred.

Therefore, it is rational to ascribe the steady high performance of Li-O₂ cell to the introduction of UH₂O₂ additive in electrolyte.

Corresponding descriptions have been made in Page 17 in manuscript.

Supplementary Fig. 8 | FTIR spectra of (i) pristine UH_2O_2 powder, (ii) G4-based electrolyte, (iii) G4- UH_2O_2 electrolyte, (iv) G4- UH_2O_2 electrolyte after discharge and (v) G4- UH_2O_2 electrolyte after recharge, respectively. In all the electrolytes, 1 M LiTFSI is dissolved.

Additionally, the article does not give a proposed mechanism for the performance-enhancing effect of H_2O_2 in a lithium-oxygen battery. Given that H_2O_2 is already present when Li_2O_2 is reacted with ambient water to form LiOH , I am skeptical as to the proposed benefit of additional H_2O_2 , as well as its stability, and thus would appreciate a proposed method of action of the H_2O_2 . I also believe a proposed mechanism, whether it is a discussion of a catalytic pathway or solubility enhancements, etc., is critical for this article to influence a wider field, as it could point to broader design principles for electrolytes in electrochemical systems. For these reasons, I do not believe this manuscript should be published in its current form.

The mechanism for the performance-enhancing effect of H_2O_2 in a lithium-oxygen battery is still under investigation. At present, our preliminary results indicate that in the presence of large amount of ambient water, the discharge products are Li_2O_2 along with intermediate LiHO_2 depositing on cathode and LiOH along with H_2O_2 dissolving in electrolyte. Upon charge, the potential can show a low value of ~ 3.5 V due to the LiHO_2 -mediated oxidation (the theoretical redox potential of LiHO_2 is calculated to be ~ 3.1 V, Chem. Commun., 2015, 51, 3189). Li_2O_2 and LiHO_2 on cathode and LiOH along with H_2O_2 in electrolyte can be reversibly decomposed. Figure Respond 1 presents our preliminary discharge-charge profiles of Li- O_2 pouch cell (solid LiSICON film is employed to protect Li metal anode) with KB cathode and electrolyte containing 30% water. In contract, when MnO_2 is added into cathode to function as a catalyst for H_2O_2 decomposition, no H_2O_2 remains in electrolyte and LiOH becomes the main discharge products. Upon charge, the potential increases to above ~ 3.8 V (Figure Respond 1). Thus, the important role of H_2O_2 is emphasized. Concerning the detailed mechanism, we are making many efforts to explore a clear understanding and it is expected to be figured out in our next work.

Figure Respond 1. Discharge-charge profiles of Li-O₂ pouch cells with KB cathode or KB-MnO₂ cathode in electrolyte containing 30% H₂O. Current density: 30 μA, 100 mA g⁻¹.

The stability of UH₂O₂ additive is evaluated through a storage experiment. One cell was firstly discharged and then rested for 7 days. Another cell was firstly performed for one discharge-charge cycle and then rested for 7 days. The open circuit voltages (OCVs) were monitored (Supplementary Fig. 7). It can be seen that during these 7 days, the OCVs shows good stability and after rest, the Li metal anodes still keep uncontaminated by H₂O, the possible decomposition product of UH₂O₂. In addition, the Li-O₂ cell performs stable discharge-charge ability for 50 cycles (Fig. 7) and there is no obvious change of Li metal anode after cycles (Fig. 4). Therefore, the stability of the UH₂O₂ additive can be confirmed.

Supplementary Fig. 7. OCVs trend of Li-O₂ cells with UH₂O₂ additive in electrolyte after a discharge process and one discharge-charge cycle, respectively. Insets are the photos of Li metal anodes after rest.

Corresponding revisions have been made in Page 13 in manuscript.

Response to Reviewers:

Reviewer #1:

Authors have addressed all the comments. I would support for the publication of this work, which brings innovation thinking to the current Li-O₂ battery research.

Thank you for your appreciation.

Reviewer #2:

The authors have done an excellent job of addressing the reviewer concerns including the new experiments that were carried out to address the concerns regarding yield and rechargeability. In light of this, I think the manuscript is now suitable for publication.

Thank you for affirming our work.

Reviewer #3:

The authors have made significant helpful revisions to their paper, but I believe there are still two points from the original reviews that have not been adequately addressed:

1) Confusion as to when LiOH is the discharge product being oxidized on charge and when Li₂O₂ is the discharge product being oxidized on charge.

My understanding is that the primary goal of this paper, according to the abstract and the concluding paragraph, is to report if adding hydrogen peroxide (and in particular an anhydrous urea-hydrogen peroxide) to a lithium-

oxygen battery can enable the reversible oxidation of lithium hydroxide. This work is seen in Figures 1-4. According to the introduction, it seems this goal is an extension of the authors' group's recent work on the promotion of LiOH as the primary discharge product by using MnO₂ and Ru in their cathode and including H₂O in their electrolyte. In addition, the authors present whether an anhydrous urea-hydrogen peroxide can serve as a helpful additive in oxidizing Li₂O₂, as seen in Figures 5-7. I think the authors still need to be clearer in both the abstract and the text (particularly introduction and conclusion) that they are discussing at times a LiOH-based lithium-oxygen electrochemistry and at times a Li₂O₂-based electrochemistry. I think both ideas are interesting and important to present; I just believe the authors should more clearly separate the two, especially as they only have evidence of a reversible, oxygen-forming oxidation reaction when Li₂O₂ is the discharge product being oxidized on charge.

Thank you for the guidance about the expression in abstract, introduction and conclusion.

As the reviewer's understanding, we initially aimed to achieve low charge potential and simultaneously avoid the utilization of noble metal Ru and transition metal compound MnO₂ in the H₂O-containing Li-O₂ battery system with LiOH as product (recently reported by our group). Thus, H₂O₂ was firstly found to efficiently facilitate the decomposition of LiOH at low charge potential. Inspired by these results, we realized the possibility of utilizing hydrogen peroxide to reduce the charge potential and improve the performance in Li-O₂ battery with Li₂O₂ as product. However, H₂O₂ generally exists in an aqueous solution and if adding the H₂O₂ aqueous solution in Li-O₂ battery with

unprotected Li metal anode, there would be severe Li metal corrosion by H₂O, largely reduced performance and safety issues (Fig. 3 in manuscript). Then, an organic urea hydrogen peroxide without H₂O contamination was developed and confirmed to play the expected role. Ultralow charge potential of ~3.26 V and good cycling stability were finally obtained.

We agree with the opinion the reviewer indicated, that the results of either the promoted decomposition of LiOH at low charge potential by the action of H₂O₂ or the reduced charge potential as well as the improved performance of Li-O₂ battery with Li₂O₂ products in the presence of organic urea hydrogen peroxide are interesting and important.

Based on our newly-added DEMS evidence of O₂ generation together with the reversible LiOH oxidation (Figure 1f-g, corresponding discussion is shown in next reply) and according to the reviewer's suggestion, we revised the related parts including abstract, introduction and conclusion in order to clearly manifest our results.

Corresponding revisions are shown below.

Abstract

Reducing the high charge potential is the crucial concern to advance the performance of Li-O₂ batteries. Here, firstly for the H₂O-containing Li-O₂ battery with LiOH products, we find that the hydrogen peroxide (H₂O₂) aqueous solution added in the electrolyte can effectively promote the decomposition of LiOH compounds at the ultralow charge potential on the catalyst-free Ketjen Black-based cathode. Furthermore, for non-aqueous Li-O₂ battery with Li₂O₂ products, we introduce an unreported urea hydrogen

peroxide chelating H_2O_2 without any H_2O in the organic as electrolyte additive in Li-O_2 battery with Li metal anode and succeed in the realization of quite low charge potential of ~ 3.26 V, which is among the best level reported. Also, the undesired H_2O generally accompanying H_2O_2 solution is circumvented to protect Li metal anode and ensure the good cycling stability of Li-O_2 battery. Our results should provide some new insights into the approaches enhancing the Li-O_2 batteries.

Introduction part (last paragraph)

In this work, for the H_2O -containing Li-O_2 battery with LiOH products, we revealed the critical role of H_2O_2 aqueous solution added into the electrolyte in assisting the decomposition of LiOH compounds at the ultralow charge potential in the case of using the common Ketjen Black carbon (KB)-based cathode. Most importantly, for non-aqueous Li-O_2 battery with Li_2O_2 products, a novel electrolyte additive urea hydrogen peroxide trapping H_2O_2 in the H_2O -free organics was first introduced to reduce the charge potential to as low as ~ 3.26 V and simultaneously avoid Li metal anode corrosion by H_2O .

Conclusion part

In summary, for the H_2O -containing Li-O_2 battery based on the LiOH compounds as products, we invented an accessible route of introducing H_2O_2 solution in the electrolyte to realize the large reduction of charge potential (from ~ 4.16 V to ~ 3.50 V). The crucial role of H_2O_2 was confirmed to greatly assist the decomposition of either solid LiOH compounds in cathode or liquid LiOH in electrolyte. Some insights may be implicated for improving the charge

ability in aqueous Li-O₂ battery. Most importantly, **for non-aqueous Li-O₂ battery with Li₂O₂ products**, we first proposed an organic H₂O₂ compound (urea hydrogen peroxide) without any H₂O as the electrolyte additive to succeed in the significant decrease of the charge potential (to ~3.26 V among the best level reported) and ...

2) The critical DEMS results to support reversible LiOH oxidation are still missing, and trying to approximate DEMS results with titration results is incorrect.

A) In our first reviews, another reviewer and I asked that differential electrochemical mass spectrometry (DEMS) be employed to verify that on charge H₂O₂ is indeed reversibly oxidizing LiOH to oxygen. Here, the authors provide titration results of extracted cathodes to show that LiOH, present before charge, is absent after charge. While this additional data is useful, the conclusion that this data can serve as a proxy for DEMS is incorrect.

Titration results and DEMS results provide two different pieces of information – the formation and decomposition of Li₂O₂ and the formation and decomposition of oxygen, respectively. For example, if more Li₂O₂ has been consumed after charge (as measured via titration) than O₂ has been evolved (as measured via DEMS), this implies that some Li₂O₂ has been oxidized to parasitic products. This idea was recently presented in studying the oxidation of LiOH in Burke et al., ACS Energy Letters, 2016, 1, 747-756. In that article, the authors looked at the reversibility of I- and H₂O-induced LiOH formation in lithium-oxygen batteries. Figure 3(c-d) show that a battery with a certain electrolyte exhibits the decomposition of nearly all LiOH on charge. While this

seems to imply reversibility, Figure 2(g-i) clearly show that no oxygen is evolved when a battery with that certain electrolyte is charged. Thus, Burke et al. shows that the absence of LiOH after charge is not sufficient evidence to claim reversible oxidation of LiOH.

Thank you for reminding us of the difference and relationship between titration and DEMS. We agree with the viewpoint that the Li_2O_2 or LiOH consumption measured via titration does not mean the corresponding evolution of O_2 from Li_2O_2 or LiOH decomposition. As the reviewer referred, although Figure 3 (c-d) in Burke et al., ACS Energy Letters, 2016, 1, 747-756, show the consumption of LiOH via titration experiment, no expected O_2 is detected in its Figure 2 (g-i) via DEMS upon charge because of the undecomposed LiO_3 formation from reaction between LiOH and I_2 . Consequently, DEMS is a convincing evidence to verify the O_2 evolution during charge and the reversibility of Li- O_2 battery. We added the DEMS measurement to further improve our work (Fig. 1f-g, see the next reply).

B) In the manuscript at hand, I still feel the authors have failed to corroborate the statement in the abstract that they, “find that the hydrogen peroxide (H_2O_2) aqueous solution added in the electrolyte can effectively promote the decomposition of LiOH compounds at the ultralow charge potential on the catalyst-free Ketjen Black-based cathode,” as the authors have indeed proven that LiOH decomposes, but have not yet proven that LiOH decomposes reversibly to oxygen, as I believe is implied.

In the rebuttal letter, the authors expressed the difficulty in measuring oxygen evolution from a cell containing hydrated H_2O_2 , as there is a rapid reaction

between water and the lithium metal anode that produces hydrogen. I understand this difficulty, and recommend that if the authors cannot perform accurate DEMS to check for O₂ evolution from LiOH during charge, that the authors note that while LiOH has been decomposed on charge, it is unclear if it has been reversibly oxidized to oxygen. I believe this article is still an important addition to the field with this acknowledgment.

Thank you for appreciating the importance of our work, even in the previous case that there was no DEMS evidence of O₂ evolution from LiOH decomposition at low charge potential in the presence of H₂O₂.

At this time, we finally fabricated a pouch cell as small as possible and successfully put it into the custom-built device for DEMS analysis for investigating whether O₂ is released during charge in the Li-O₂ pouch cell with in situ formed LiOH/KB cathode and H₂O₂-containing electrolyte. As the pouch cell was introduced in the main text (Page 5), “on the anode side, the Li metal is protected by utilizing the Li ion conducting glass-ceramic film (LiSICON, Li₂O-Al₂O₃-SiO₂-P₂O₅-TiO₂-GeO₂, Ohara Corporation), only allowing the transport of Li⁺ and preventing the penetration of others. On the cathode side, a KB-based cathode and a glass fiber infiltrating electrolyte are constructed”. In this case, the rapid reaction between water and the lithium metal anode during DEMS measurement can be avoided. Figure 1f-g shows the DEMS results.

It can be seen that obvious O₂ evolution was detected at ~3.50 V during charge process and there was no evolution of CO₂. Combined with the previous titration results, we confirmed LiOH could decompose reversibly to O₂. Then, our statement in the abstract “find that the hydrogen peroxide (H₂O₂)

aqueous solution added in the electrolyte can effectively promote the decomposition of LiOH compounds at the ultralow charge potential on the catalyst-free Ketjen Black-based cathode,” should be rational.

Figure 1 (f) Charge profiles of Li-O₂ pouch cell with in situ formed solid LiOH/KB cathode and G4-H₂O-H₂O₂ electrolyte during the in situ DEMS (differential quantitative mass spectrometry) measurement. (g) Corresponding O₂ and CO₂ evolution rate during charge.

3) The DEMS results presented in Figure 6 do indeed show reasonable reversibility of Li₂O₂ in a cell containing urea-H₂O₂, and I appreciate this addition.

Thank you for appreciating our effort.

Response to Reviewers:

Reviewer #3:

The authors have made the requested revisions, delineating between the LiOH-based electrochemistry and the Li₂O₂-based electrochemistry and adding the DEMS work for the LiOH-based electrochemistry. I agree that the manuscript is now suitable for publication.

Thank you for your appreciation.